



# Early Cenozoic Eurekan strain partitioning and decoupling in central Spitsbergen, Svalbard

**Jean-Baptiste P. Koehl**[1,2,3,4]

[1]Faculty of Environmental Sciences and Natural Resource Management, Norwegian University of Life Sciences, P.O. Box 5003 NMBU, NO-1432 Ås, Norway.

[2]Department of Geosciences, UiT The Arctic University of Norway in Tromsø, NO-9037 Tromsø, Norway.

[3]Research Centre for Arctic Petroleum Exploration (ARCEx), University of Tromsø, NO-9037 Tromsø, Norway.

[4] CAGE – Centre for Arctic Gas Hydrate, Environment and Climate, NO-9037 Tromsø, Norway.

**Correspondence:** Jean-Baptiste P. Koehl (jean-baptiste.koehl@uit.no)

## Abstract

The present study of field, petrological, exploration well and seismic data shows that backward-
dipping duplexes comprised of phyllitic coal and bedding-parallel décollements and thrusts, which localized along lithological transitions in tectonically thickened Lower–lowermost Upper Devonian, uppermost Devonian–Mississippian and uppermost Pennsylvanian–lowermost Permian sedimentary strata of the Wood Bay and/or Widje Bay and/or Grey Hoek formations, of the Billefjorden Group and of the Wordiekammen Formation respectively, partially decoupled
uppermost Devonian–Permian sedimentary rocks of the Billefjorden and Gipsdalen groups from Lower–lowermost Upper Devonian rocks of the Andrée Land Group and Mimerdalen Subgroup during early Cenozoic Eurekan deformation in central Spitsbergen. Eurekan strain decoupling along these structures explains differential deformation between Lower–lowermost Upper Devonian rocks of the Andrée Land Group/Mimerdalen Subgroup and overlying uppermost
Devonian–Permian sedimentary strata of the Billefjorden and Gipsdalen groups in central–northern Spitsbergen without requiring an episode of (Ellesmerian) contraction in the Late Devonian. Potential formation mechanisms for bedding-parallel décollements and thrusts include shortcut faulting, and/or formation as a roof décollement in a fault-bend hanging wall (or ramp) anticline, as an imbricate fan, as an antiformal thrust stack, and/or as fault-propagation folds over
reactivated/overprinted basement-seated faults. The interpretation of seismic data in Reindalspasset indicates that Devonian sedimentary rocks of the Andrée Land Group and





Mimerdalen Subgroup might be preserved east of the Billefjorden Fault Zone, suggesting that the Billefjorden Fault Zone did not accommodate reverse movement in the Late Devonian. Hence, the thrusting of Proterozoic basement rocks over Lower Devonian sedimentary rocks along the

Balliolbreen Fault and fold structures within strata of the Andrée Land Group and Mimerdalen Subgroup in central Spitsbergen may be explained by a combination of down-east Carboniferous normal faulting with associated footwall rotation and exhumation, and subsequent top-west early Cenozoic Eurekan thrusting along the Billefjorden Fault Zone. Finally, the study shows that major east-dipping faults, like the Billefjorden Fault Zone, may consists of several, discrete, unconnected

(soft-linked and/or stepping) or, most probably, offset fault segments that were reactivated/overprinted with varying degree during Eurekan deformation due to strain partitioning and/or decoupling along sub-orthogonal NNE-dipping reverse faults.

## 1. Introduction

The main goal of this contribution is to examine the influence of strain decoupling and partitioning on deformation patterns within Devonian–Permian sedimentary successions in central Spitsbergen during the early Cenozoic Eurekan tectonic event. The impact of this event, though well studied in western Spitsbergen where it resulted in the formation of the West Spitsbergen Fold-and-Thrust Belt (Dallmann et al., 1988, 1993; Braathen et al., 1999) with multiple levels of

detachment and décollement (Maher, 1984; Maher et al., 1986; Bergh et al., 2000), lacks detailed characterization in central Spitsbergen (Figure 1a; see DataverseNO for high-resolution versions of all figures and supplements https://doi.org/10.18710/MXKQPE).

The study discusses the presence of bedding-parallel décollement levels and imbricate link thrusts (McClay and Insley, 1986) arranged into gently dipping duplexes within weak sedimentary

beds of the Andrée Land Group, Billefjorden Group and Wordiekammen Formation, and their role in partially decoupling Eurekan deformation in late Paleozoic sedimentary successions. Potential formation mechanisms, such as shortcut faulting (Buiter and Pfiffner, 2003), and the influence of preexisting inherited structures (e.g., Billefjorden Fault Zone) are reviewed.

The study also briefly discusses implications for the Ellesmerian Orogeny, a poorly

constrained short-lived episode of contractional deformation in the Late Devonian that presumably explains the juxtaposition of Proterozoic basement against Lower–lowermost Upper Devonian sedimentary strata of the Andrée Land Group and Mimerdalen Subgroup (Vogt, 1938; Harland et



al., 1974; McCann, 2000; Piepjohn, 2000; Piepjohn et al., 2000; Piepjohn and Dallmann, 2014), and differential deformation between folded Devonian rocks of the Andrée Land Group and

Mimerdalen Subgroup and poorly deformed rocks of the uppermost Devonian–Permian Billefjorden and Gipsdalen groups in Dickson Land in central Spitsbergen. Notably, the contribution shows that Eurekan deformation localized in weak, intensely deformed sedimentary units of the uppermost Devonian–Permian sedimentary rocks of the Billefjorden and Gipsdalen rocks, that Devonian sedimentary rocks of the Andrée Land Group are possibly preserved east of

the Billefjorden Fault Zone and, thus, that the Billefjorden Fault Zone most likely did not act as a reverse fault in the Late Devonian, and that juxtaposition of Proterozoic basement against Lower Devonian rocks in central Spitsbergen may be achieved through Carboniferous normal faulting and early Cenozoic Eurekan top-west thrusting.

Finally, the study considers the significant along-strike variations in geometry and

kinematics of the Billefjorden Fault Zone, and discusses the extent and potential segmentation of this fault in conjunction with a new trend of NNE-dipping faults striking suborthogonal to the main N–S-trending grain in the study area. The role of these suborthogonal faults in Eurekan strain partitioning is briefly discussed.

**2. Geological setting**

*2.1. Caledonian Orogeny*

Spitsbergen is composed of three terranes that started assembling during the Caledonian Orogeny and were juxtaposed against one another by N–S-striking crustal faults like the Billefjorden Fault Zone (Harland and Wright, 1979; Ohta et al., 1989, 1995; Gee and Page, 1994).

Caledonian deformation was accompanied by tectonothermal events with high-grade (eclogite and blueschist) metamorphism from mid-Cambrian to late Silurian times that occurred during subduction and closure of the Iapetus Ocean and that are partly preserved in northwestern (Ohta et al., 1989) and western Spitsbergen (Horsfield, 1972; Kośmińska et al., 2014).

Caledonian grain in western, northwestern, central and eastern Spitsbergen forms major,

gently plunging, N–S-trending folds and thrust stacks with well-developed foliation, e.g., the Atomfjella Antiform in Ny-Friesland (Figure 1b), an antiformal thrust stack that consists of a succession of nappes composed of Proterozoic granite and metasedimentary rocks separated by



west-verging (Flood et al., 1969; Balashov et al., 1993; Witt-Nilsson et al., 1998; Johansson and Gee, 1999; Johansson et al., 2004, 2005) and/or top-east thrusts (Manby and Michalski, 2014).


### 2.2. Devonian late–post-orogenic collapse

In the Early Devonian, late–post-Caledonian gravitational collapse initiated (Chorowicz, 1992; Roy, 2007, 2009; Roy et al., unpublished) leading to the deposition of several km-thick (Old Red Sandstone) basins throughout Spitsbergen (Birkenmajer and Turnau, 1962; Harland et al., 1974; Manby and Lyberis, 1992; Manby et al., 1994; Dallmann and Piepjohn, 2020) and emplacement of late-orogenic plutons in northwestern, central and eastern Spitsbergen (Hamilton et al., 1962; Gayer et al., 1966; Ohta et al., 2002; Myhre et al, 2008).

In northern Spitsbergen, Devonian sedimentary rocks of the Siktefjellet, Red Bay and Andrée Land groups (Gee and Moody-Stuart, 1966; Friend et al., 1966; Friend and Moody-Stuart, 1972; Murascov and Mokin, 1979; Friend et al., 1997) deposited during extension and subsidence along N–S-striking normal faults, forming west-tilted (half-) grabens, e.g., in Raudfjorden, Bockfjorden (Manby and Lyberis, 1992; Manby et al., 1994), Andrée Land and Kota (Roy, 2007, 2009; Roy et al., unpublished; Figure 1a). However, other works argue that Devonian sedimentary deposits of the Andrée Land Group and Mimerdalen Subgroup in Svalbard deposited along low-angle, post-Caledonian detachments that accommodated large amounts of top-east, normal movement (e.g., the Woodfjorden detachment) and are associated with syn-kinematic east-verging folds (Roy, 2007, 2009; Roy et al., unpublished). In addition, recent studies show that basement ridges, e.g., the Bockfjorden Anticline in northwestern Spitsbergen, may have exhumed as core complexes along low-angle extensional detachments (e.g., the Keisarhjelmen detachment), and K–Ar geochronology suggests that exhumation occurred from late Silurian to Late Devonian times (Braathen et al., 2018).

### 2.3. Ellesmerian Orogeny

Ellesmerian deformation is thought to have initiated in the Late Devonian–Early Mississippian, possibly in the Late Frasnian–Famennian (Vigran, 1964; Allen, 1965, 1973; Pcelina et al., 1986; Brinkmann, 1997; Schweitzer, 1999; Piepjohn et al., 2000) and was presumably recorded by the deposition of coarse-grained sedimentary rocks of the Mimerdalen Subgroup (Planteryggen and Plantekløfta formations; Piepjohn and Dallmann, 2014). However, recent fossil



and spore analysis suggest an early Frasnian (ca. 380 Ma) age for these stratigraphic units (Berry
and Marshall, 2015). Deformation is believed to have stopped prior to the deposition of middle–
late Famennian–Mississippian (Scheibner et al., 2012; Lindemann et al., 2013; Marshall et al.,
2015; Würtzen et al., 2019; Lopes, pers. comm. 2019) sedimentary rocks of the Billefjorden Group
(Vogt, 1938; Piepjohn, 2000). Previous works also suggested that hundreds–thousands of
kilometer-scale strike-slip movement along N–S-striking faults, e.g., Billefjorden Fault Zone,
finalized the accretion of basement terranes constituting the Svalbard Archipelago (Harland et al.,
1974; Harland and Wright, 1979; Ohta et al., 1989), while more recent studies argue for limited
amounts of strike-slip movement (McCann, 2000; Piepjohn, 2000).

In Pyramiden, in Dickson Land (northern–central Spitsbergen; Figure 1b), Proterozoic
basement rocks were thrusted top-west onto Lower Devonian sedimentary rocks of the Wood Bay
Formation along the Balliolbreen Fault (Harland et al., 1974; Piepjohn, 2000; Bergh et al., 2011)
in Late Devonian times, and presumably undeformed uppermost Devonian–Mississippian clastic
and coal-bearing sedimentary deposits of the Billefjorden Group overlie folded Lower–lowermost
Upper Devonian metasedimentary rocks that were involved in Ellesmerian deformation. In
Triungen (Figure 1a–b), folded–gently dipping Lower Devonian rocks of the Wood Bay Formation
are juxtaposed against flat-lying, undeformed, uppermost Devonian–Permian strata of the
Billefjorden Group and Wordiekammen Formation along the Triungen–Grønhorgdalen Fault Zone
(McCann and Dallmann, 1996). In Sentinelfjellet and Odellfjellet (Figure 1b), the Balliolbreen
Fault thrusted Proterozoic basement rocks in the hanging wall over Devonian sedimentary rocks
of the Andrée Land Group and Mimerdalen Subgroup in the footwall and is thought to be
unconformably overlain by undeformed, uppermost Devonian–Mississippian sedimentary rocks of
the Billefjorden Group, thus suggesting Late Devonian top-west thrusting (Friend and Moody-
Stuart, 1972; Harland et al., 1974; Lamar et al., 1986).

### 2.4. Carboniferous basins

In Carboniferous times, ENE–WSW extension formed narrow, kilometer- to tens of
kilometer-wide, N–S- to NW–SE-trending troughs, e.g., Billefjorden Trough (Maher Jr., 1996;
McCann and Dallmann, 1996; Braathen et al., 2011), bounded by major faults such as the
Billefjorden Fault Zone (Harland et al., 1974), which was reactivated as a normal fault from
Odellfjellet in the north to Reindalspasset in the south (Bælum and Braathen, 2012; Figure 1a–b).



Shortly after the end of Ellesmerian deformation, partly eroded Devonian sedimentary rocks of the Andrée Land Group and Mimerdalen Subgroup were covered by uppermost Devonian–Mississippian (Marshall et al., 2015), fluvial, coal- and clastic-rich deposits of the Billefjorden Group (Cutbill and Challinor, 1965; Cutbill et al., 1976; Aakvik, 1981; Gjelberg, 1981, 1984). These are divided into the Hørbyebreen and Mumien formations, which are composed of the

Triungen and Hoelbreen, and Sporehøgda and Birger Johnsonfjellet members respectively. The Triungen and Sporehøgda members dominantly consist of clastics whereas the Hoelbreen and Birger Johnsonfjellet members are composed of coal seams and coaly shales (Cutbill and Challinor, 1965; Cutbill et al., 1976; Aakvik, 1981; Gjelberg and Steel, 1981; Gjelberg, 1984).

These deposits are found in Arctic areas stretching from the Barents Sea (Bugge et al.,

1995; Larssen et al., 2002) to Arctic Canada (Emma Fiord Formation; Davies and Nassichuck, 1988) and were presumably deposited during a period of tectonic quiescence (Johannessen and Steel, 1992; Braathen et al., 2011; Smyrak-Sikora et al., 2018), though a syn-tectonic deposition was also proposed for these rocks in Arctic Canada (Beauchamp et al., 2018), the Barents Sea (Koehl et al., 2018), Bjørnøya (Gjelberg, 1981), and in Spitsbergen in the northern part of the

Billefjorden Trough (Koehl and Muñoz-Barrera, 2018).

In the Pennsylvanian, fluvial to shallow marine sedimentary strata of the Gipsdalen Group were deposited in subsiding basins. These are divided into the Hultberget, Ebbadalen, Minkinfjellet, Wordiekammen and Gipshuken formations in central Spitsbergen (Cutbill and Challinor, 1965; Johannessen, 1980; Gjelberg and Steel, 1981; Johannessen and Steel, 1992;

Braathen et al., 2011; Smyrak-Sikora et al., 2018), all of which range from late Serpukhovian to earliest Permian in age.

Sedimentary strata of the Gipsdalen Group are mostly composed of clastic, carbonate and evaporitic deposits and karst breccia, and represent the thickest sedimentary succession in the Billefjorden Trough (McWhae, 1953; Cutbill and Challinor, 1965; Holliday and Cutbill, 1972;

Johannessen, 1980; Lønøy, 1995). The deposition of sedimentary strata of the Hultberget, Ebbadalen, and Minkinfjellet formations was accompanied by kilometer scale normal displacement along N–S-striking faults like the Billefjorden Fault Zone, whereas the Wordiekammen and Gipshuken formations were deposited during minor tectonic activity (Gjelberg and Steel, 1981; Fedorowski, 1982; Braathen et al., 2011; Smyrak-Sikora et al., 2018).






*2.5. Eurekan deformation*

In the Paleocene (ca. 62 Ma), Eurekan deformation initiated in western Spitsbergen due to the opening of the Labrador Sea and Baffin Bay between Canada and Greenland (Chalmers and Pulvertaft, 2001; Oakey and Chalmers, 2012) and resulted in the formation of the West Spitsbergen

Fold-and-Thrust Belt between Kongsfjorden and Sørkapp (Harland, 1969; Lowell, 1972; Harland and Horsfield, 1974; Maher et al., 1986; Dallmann et al., 1988, 1993; Andresen et al., 1994; Bergh and Grogan, 2003) and formation of a foreland basin, the Tertiary Central Basin, in central Spitsbergen (Larsen, 1988; Petersen et al., 2016). Eurekan thrusts and folds in Spitsbergen dominantly strike and trend NNW–SSE (Harland and Horsfield, 1974; Bergh and Andresen, 1990;

Dallmann et al., 1993; Bergh et al., 2011; Blinova et al., 2012) except in Kongsfjorden (Figure 1a) where they strike and trend WNW–ESE (Bergh and Andresen, 1990; Bergh et al., 2000; Saalmann and Thiedig, 2000, 2001; Piepjohn et al., 2001). Early Cenozoic thrusts in western Spitsbergen commonly form décollements in shaly beds, e.g., in Triassic shales in Midterhuken (Maher, 1984; Maher et al., 1986; Figure 1a). In central–eastern Spitsbergen, major N–S-striking brittle faults like

the Billefjorden Fault Zone were partly reactivated by Eurekan deformation in Flowerdalen (Harland et al., 1974; Haremo et al., 1990; Haremo and Andresen, 1992; Figure 1b) and Reindalspasset (Bælum and Braathen, 2012) in the south, but were apparently unaffected in nothern areas like Sentinelfjellet (Figure 1b) where uppermost Devonian–Mississippian strata seem to unconformably lie over the fault (Harland et al., 1974).


## 3. Methods

*3.1. Seismic, field and petrological data, and satellite images*

The present contribution uses structural measurements of bedding and fracture surfaces in Devonian–Mississippian sedimentary strata collected in summer 2016 in Pyramiden (Figure 1b).

The study also uses microscopic analysis of fault rocks and sedimentary rocks adjacent to brittle faults as a confirmation tool (included in supplement 1).

Seismic data in nearshore fjords in central Spitsbergen are from the Norwegian Petroleum Directorate and uninterpreted seismic lines are provided in supplement 2. Seismic interpretation was tied to data from exploration well 7816/12-1 in Reindalspasset (Figure 1a–b; Eide et al., 1991)

and time–depth conversion of well data is based on checkshots from Equinor and Store Norske Spitsbergen Kulkompani. The well penetrated late Paleozoic–Mesozoic sedimentary rocks and





ends at a depth of 2261 m with 54 m of uppermost Devonian–Mississippian strata of the Billefjorden Group.

**4. Results**

*4.1. Field and petrological data*

*4.1.1. Pyramiden*

In Pyramiden, a steeply east-dipping, N–S-striking brittle fault crops out in a gully below the entrance of the Russian coal mine (Figure 2). This fault is located half-way to the mine in the

gully and crosscuts steeply east-dipping Lower Devonian sedimentary rocks of the Wood Bay Formation, which are involved into a large fold structure with Devonian bedding surfaces locally overturned to the east (Figure 2 and Figure *3*a, and supplement 3). The fault shows meter-thick lenses of cataclastic fault rock (supplement 1). Devonian sedimentary rocks are dominated by poorly deformed quartz crystals showing undulose extinction and limited recrystallization

(supplement 1), whereas cataclastic fault rock shows distributed fractures with little (centimeter-scale) to no displacement.

There is no trace of Proterozoic basement rocks in this area although field studies and geological maps suggest that Proterozoic basement was thrusted over Lower Devonian strata along the Balliolbreen Fault (McCann, 1993; McCann and Dallmann, 1996; Piepjohn et al., 1997;

Dallmann et al., 1999, 2004; Bergh et al., 2011; svalbardkartet.npolar.no). Sample preparation for thin sectioning actually proved problematic for Devonian sedimentary rocks located in the hanging wall of the presumed fault, which resulted in misleading thick sections showing quartz crystals resembling pyroxenes (supplement 1). Thus, it is more likely that earlier maps showing exclusively Devonian–Mississippian sedimentary rocks of the Wood Bay Formation and Billefjorden Group

below the mine entrance by Harland et al. (1974), Aakvik (1981), Lamar et al. (1986), and Trust Arktikugol (1988; Sirotkin, pers. comm. 2019) are correct.

Farther up the gully, a one–two meter-thick succession of interbedded sandstone and coal is juxtaposed against steeply east-dipping Lower Devonian strata to the west and overlain by a (at least three meter) thick layer of uppermost Devonian–Mississippian coals of the Billefjorden Group

that shows phyllitic shear fabrics (Figure 2 and Figure *3*b and supplement 4). The presence of abundant coal suggests that this one–two meters thick unit is part of the Billefjorden Group as well. Bedding surfaces within the one–two meter-thick succession dip gently–steeply to the east (Figure





3a), display sigmoidal geometries with Z-like shapes, and terminate abruptly against the three

meter-thick layer of uppermost Devonian–Mississippian phyllitic coal upwards and against Lower

Devonian rocks downwards (dashed yellow lines in Figure 3b). In addition, coaly shales within

this succession display phyllitic fabrics similar to those observed within overlying coals, and seem

to form repeated successions of alternating beds of sandstone and coaly shale truncated by steeply

east-dipping sigmoidal fault surfaces (thin dashed red lines in Figure 3b). The Z-like sigmoidal

shape of bedding surfaces, phyllitic shear fabrics of the coaly shales, and possible repetitions of the

succession suggest that the steeply east-dipping, sigmoidal faults crosscutting the succession are

imbricate thrust faults (stereonet 3 in Figure 2), i.e., possible link thrusts (McClay and Insley,

1986), which accommodated top-west to top-WNW movements. The truncation of sandstone–

coaly shale beds upwards and downwards, the abrupt transition (partly covered by screes) with

underlying Lower Devonian rocks and overlying uppermost Devonian–Mississippian coals, and Z-

shaped phyllitic shear fabrics within overlying coals suggest that the sandstone–coaly shale

succession is bounded by moderate–low-angle, east-dipping floor- and roof-thrusts (McClay,

1992) with top-west to top-WNW sense of shear. In cross-section, the interaction of intra-

succession, steeply east-dipping link thrusts and inter-succession, moderate–low-angle floor- and

roof-thrusts defines an east-dipping duplex structure (Boyer and Elliott, 1982) of imbricate thrusts

bounded upwards and downwards by potential décollements and/or detachments parallel to original

(i.e., prior to deformation) bedding surfaces (e.g., thick red lines showing the transition from

interbedded coaly shales and sandstone to coal, and from coal to sandstone in Figure 3b). The

nomenclature of hindward/forward-dipping duplexes of Boyer and Elliott (1982) does not apply

here since the foreland of the West Spitsbergen Fold-and-Thrust Belt (Tertiary Central Basin) is

located southeast of Pyramiden. Thus, the term "backward" is used to describe the east-dipping

character of the duplexes, i.e., oppositely to the inferred transport direction.

Above the mine entrance, sedimentary rocks of the Billefjorden Group are dominated by

yellow sandstone that are crosscut by dominant WNW–ESE-striking fractures and subsidiary N–

S- and ENE–WSW-striking fractures (stereonets 1 and 2 in Figure 2) showing oblique-slip

kinematics. Poorly preserved slickenside lineations did not yield any information on relative

displacement between footwall and hanging wall. In the west, dark sandstone and quartzite crop

out and contain fossil wood, which are probably Lower Devonian in age. The contact between the

Lower (–lowermost Upper?) Devonian dark sandstone and uppermost Devonian–Mississippian



yellow sandstone of the Billefjorden Group, and intra-Devonian lithological contacts (e.g., between

Devonian quartzite and dark sandstone; Figure 3a), although partly covered by screes and/or mostly

made of loose blocks, do not appear to be faulted or tectonized and trend c. WNW–ESE to NW–

SE as bedding surfaces appear to change from moderately–steeply east-dipping below the mine

entrance to gently NNE-dipping above the mine entrance (Figure 2 and Figure _3_a), i.e., parallel to

the dominant fault trend in both uppermost Devonian–Mississippian (stereonet 1 in Figure 2) and

Lower (–lowermost Upper?) Devonian rocks (stereonet 4 in Figure 2).

Noteworthy, most outcrops of uppermost Devonian–Mississippian strata in this part of the

study area trend E–W to WNW–ESE. Thus, the dominance of WNW–ESE-striking faults is

unlikely the result of measurements flawed by a preferential outcrop trend, since E–W- to WNW–

ESE-trending outcrops would rather favor identification and measurement of N–S-striking faults.

A possible interpretation of outcrops and structures in Pyramiden (Figure 1b) is that the

subvertical, N–S-striking brittle fault within steeply east-dipping Lower Devonian strata in the

gully below the coal mine entrance (Figure 2 and Figure _3_a) represents the Balliolbreen Fault

segment of the Billefjorden Fault Zone, and that low-angle roof/floor thrusts between Lower (–

lowermost Upper?) Devonian rocks and the overlying succession of uppermost Devonian–

Mississippian sandstone, coaly shale and coal (Figure 3b) correspond to the upward-flattening

continuation of this fault. However, no fault was observed between Lower (–lowermost Upper?)

Devonian rocks of the Andrée Land Group (and Mimerdalen Subgroup) and sandstones of the

Billefjorden Group above the mine, and lithogical and stratigraphic contacts there display

significantly different trends (WNW–ESE to NW–SE; Figure 3a).


### 4.1.2. Triungen

Fieldwork in Triungen (see location in Figure 1a–b) shows that the Triungen–

Grønhorgdalen Fault Zone (McCann and Dallmann, 1996) and the contact between Lower

Devonian of the Wood Bay Formation and overlying uppermost Devonian–Mississippian

sedimentary rocks of the Billefjorden Group along the fault are largely covered by dark screes

(Figure 3c–e). In the hanging wall though, Lower Devonian sedimentary strata are moderately

tilted to the south and define an angular unconformity with overlying, flat-lying strata of the

Billefjorden Group (Figure 3c). Based on the presence of thick, flat-lying, coal-rich strata in the

lower part of the Billefjorden Group overlying Lower Devonian sedimentary strata in the hanging





wall of the fault, the dark screes along the fault trace (Figure 3d–e) are believed to represent
        uppermost Devonian–Mississippian coals–coaly shales that might have been dragged along the
        Triungen–Grønhorgdalen Fault Zone during tectonic movements.

### 4.2. Seismic data

#### 4.2.1. Seismic units and stratigraphy

        In seismic sections, Precambrian–Caledonian basement rocks commonly show chaotic
        reflections, most likely arising from their complex tectonic history (e.g., Caledonian folding,
        shearing, thrusting and post-Caledonian extensional and contractional overprints), and subparallel
        reflections, possibly corresponding to seismic artifacts (e.g., multiples; Figure 4a–g; see
DataverseNO for high-resolution versions of all figures and supplements;
        https://doi.org/10.18710/MXKQPE).

        Potential Devonian rocks of the Andrée Land Group in Reindalspasset (Figure 1a–b) are
        characterized by partly disrupted, semi-continuous, sub-parallel to chaotic, moderate- to low-
        amplitude seismic reflections (Figure 4g). The moderate- to low-amplitude character of internal
seismic reflections within this seismic unit suggests that it is made up with relatively homogeneous
        deposits with minor lithological variations. Thus, Devonian rocks in Reindalspasset are likely
        composed of thick successions of medium- to fine-grained sedimentary rocks such as siltstone and
        shales, possibly of the Lower– Devonian Wood Bay (or time-equivalent Marietoppen Formation
        in southern Spitsbergen) and/or Middle Devonian Grey Hoek and/or Wijde Bay formations.

Uppermost Devonian–Mississippian sedimentary rocks are characterized by high-
        amplitude seismic reflections that are most likely the product of acoustic impedance contrast
        between low density coal seams interbedded with clastic deposits. Such seismic facies is relatively
        common for uppermost Devonian–Mississippian sedimentary rocks in the Norwegian Barents Sea
        (Koehl et al., 2018; Tonstad, 2018). In Reindalspasset, uppermost Devonian–Mississippian,
phyllitic, coal-rich deposits of the Billefjorden Group were penetrated by exploration well 7816/12-
        1 at a depth of 2261 m (Eide et al., 1991), which corresponds to a time of 0.96 s (TWT) when time-
        converted (Figure 4g).

        Pennsylvanian–Permian sedimentary strata of the Gipsdalen Group are mostly composed
        of packages of subparallel low- to moderate-amplitude seismic reflections separated by discrete,
moderate- to high-amplitude reflections. The Hultberget and Ebbadalen formations dominantly





show partly disrupted, subparallel reflections possibly representing medium- to fine-grained sedimentary strata (e.g., of the Trikolorfjellet Member) that, in places, alternate with chaotic seismic facies probably characterizing coarse-grained sedimentary deposits (e.g., of the Odellfjellet and/or Ebbaelva members; Johannessen, 1980; Johannessen and Steel, 1992; Braathen et al., 2011;

Smyrak-Sikora et al., 2018). The Minkinfjellet and Wordiekammen formations are dominated by a thick package of sub-parallel, moderate- to low-amplitude seismic reflections mostly representing carbonate and gypsum deposits (Figure 4). The top reflection of the Wordiekammen Formation is characterized by high amplitude and is relatively easy to trace throughout the study area (Figure 4). Finally, the Gipshuken Formation displays chaotic to subhorizontal and subparallel low-

amplitude seismic reflections (Figure 4). The Wordiekammen and Gipshuken formations are easily identified on seismic data because they crop out at sea level along the northern shore of Sassenfjorden and Tempelfjorden and, hence, can be directly tied to onshore geology (Dallmann et al., 2004, 2009; Dallmann, 2015). Mesozoic sedimentary rocks are not the focus of the present study and were therefore not described.


### 4.2.2. *Structures in Sassenfjorden–Tempelfjorden*

Seismic data in Sassenfjorden–Tempelfjorden (Figure 1a–b) show that basement rocks and overlying, uppermost Devonian–Permian sedimentary rocks of the Billefjorden and Gipsdalen groups are folded into two open, upright, NW–SE- to WNW–ESE-trending fold structures that

coincide with similarly trending, several kilometer-wide, elongated ridges representing uplifted portion of the seafloor in Sassenfjorden and Billefjorden (Koehl, 2020; Koehl et al., submitted), and with steeply NNE-dipping, basement-seated faults mostly confined to basement (–Devonian?) rocks and uppermost Devonian–Mississippian coal-rich deposits of the Billefjorden Group, or that die out upwards in the lower part of the Gipsdalen Group (Figure 4a). Based on the minor reverse,

top-SSW offset of thickened uppermost Devonian–Mississippian sedimentary strata, it is probable that the two gentle fold structures formed in the early Cenozoic as fault-propagation folds due to upward propagation and reverse reactivation/overprinting of NNE-dipping basement-seated faults.

Seismic data in Sassenfjorden and Tempelfjorden also show that high-amplitude seismic reflections characterizing uppermost Devonian–Mississippian sedimentary rocks significantly

thicken (approximately twice thicker) towards the south-southwest, near the intersection of the east-dipping Billefjorden Fault Zone with NNE-dipping basement-seated faults, potentially





suggesting that uppermost Devonian–Mississippian rocks represent early syn-rift sedimentary deposits (Prosser, 1993) and are part of the initiation stage (Gawthorpe and Leeder, 2000) of the Billefjorden Trough (Figure 4a–b). There, high-amplitude seismic reflections representing coal-

rich uppermost Devonian–Mississippian strata display laterally disrupted, (SSW-) tilted, Z-shaped geometries (Figure 4b and e) that contrast with continuous, subparallel, subhorizontal geometries of the reflections in the northeast (Figure 4a, and c–e). Since similar Z-shaped geometries interpreted as duplex structures comprised of bedding-parallel décollements (floor- and roof-thrusts) connected by bedding-oblique link-thrusts were encountered in locally thickened, coal-

rich, uppermost Devonian–Mississippian sedimentary deposits in Pyramiden (Figure 3b), it is conceivable that, in Sassenfjorden–Tempelfjorden too, significant rheological contrasts between uppermost Devonian–Mississippian coal–coaly shale and sandstone of the Billefjorden Group localized the formation of duplex-related décollements and thrust faults during early Cenozoic deformation.

Locally, moderate- to low-amplitude, subparallel seismic reflections of the Hultberget, Ebbadalen, Minkinfjellet and Wordiekammen formations are disrupted by and slightly bending along moderate to shallow dipping, bedding-oblique reflections, which are interpreted as minor early Cenozoic thrust faults (Figure 4a, c, d and f). These minor thrusts appear to flatten downwards and die out within high-amplitude seismic reflections of the Billefjorden Group, thus supporting

the presence of bedding-parallel décollements in uppermost Devonian–Mississippian sedimentary rocks (Figure 4c).

Seismic reflections within the overlying Gipshuken Formation dip gently to moderately and display continuous to partly chaotic facies (Figure 4f). These are disrupted by possible gently NE- to east- and SW- to west-dipping thrusts that seem to flatten downwards into the Top

Wordiekammen Formation reflection, forming part of possible imbricate thrust systems (Figure 4a, c, d and f) resembling thrusts within coals and coaly shales of the Billefjorden Group (Figure 3b). This interpretation is supported by onshore Eurekan thrust geometries on the northern shore of Sassenfjorden (supplement 5). This suggests the presence of (a) décollement level(s) within the Wordiekammen Formation and/or at the boundary between the Wordiekammen and Gipshuken

formations. Internal seismic packages display significant thickness variations, pinching out laterally and, in places, becoming as thick as the whole Gipshuken Formation (Figure 4a, c, d and f). These thickness variations are tentatively related to tectonic thickening due to early Cenozoic





thrusting and, potentially, to the presence of partially mobile evaporite within the Gipshuken
Formation (Dallmann et al., 1999).


### 4.2.3. Structures in Reindalspasset

Seismic data in Reindalspasset show a N–S-trending open fold structure (Figure 4g). In
Lower–Middle Devonian rocks, the lowermost part of the fold shows semi-continuous to chaotic,
moderate- to low-amplitude, locally undulating seismic reflections that display intensive disruption

and wedge-shaped geometries (Figure 4g). Moderate- to low-amplitude reflections within wedge-
shaped seismic packages display S- and Z-shaped geometries that are disrupted respectively by
moderately west- and east-dipping reflections that appear to be responsible for the thickening of
internal units and that flatten and die out upwards prior to or at the boundary with overlying
uppermost Devonian–Mississippian rocks (Figure 4g). These wedge-shaped seismic packages are

interpreted as thickened sheets crosscut by early Cenozoic thrust faults that, in places, form duplex
structures comprised of floor- and roof-thrusts connected by link thrusts. Associated undulating
reflection geometries are thought to represent folding. Based on the sub-continuous, low- to
moderate-amplitude seismic facies and on the presence of folds and bedding-subparallel thrusts, it
is probable that (at least the upper part of) this seismic unit is composed of shale-rich, Lower–

Middle Devonian sedimentary strata of the Wood Bay and/or Grey Hoek and/or Wijde formations.

The core of the fold is partly composed of gently west-dipping to flat-lying, high-amplitude
seismic reflections representing coal-rich sedimentary strata of the Billefjorden Group, which were
penetrated by exploration well 7816/12-1 at a depth of 2261 m (Eide et al., 1991), i.e., 0.96 s (TWT;
Figure 4g). In the east, sedimentary strata of the Billefjorden Group can be traced as continuous,

gently west-dipping, sub-parallel reflections that thicken westwards against the eastern limb of the
fold and that are locally folded and disrupted by a few gently west-dipping, bedding-subparallel
reflections that accommodate local thickening of the Billefjorden Group and, hence, may represent
minor early Cenozoic thrust faults (Figure 4g). High-amplitude reflections of the Billefjorden
Group are thickest within the fold hinge, where they show undulating geometries and are

intensively disrupted. These disruptions may be the result of early Cenozoic thrusting along low-
angle, bedding-subparallel faults, which are probably responsible for the thickening of uppermost
Devonian–Mississippian strata within the fold hinge and are possibly forming part of an antiformal
stack or ramp anticline (Figure 4g). The largest of these potential early Cenozoic thrusts localized



along the boundary between uppermost Devonian–Mississippian and Pennsylvanian sedimentary
strata, i.e., parallel to the eastern limb of the fold, and splays upwards into four faults. This fault
and associated splays quickly die out upwards within the fold hinge in the upper part of the
uppermost Devonian–Mississippian and in the lower part of the Pennsylvanian sedimentary
succession, offset sediments of the Billefjorden and Gipsdalen groups in a reverse manner (possible
repeated portion of the Billefjorden Group), and flatten into the base of the Billefjorden Group or
uppermost part of the Lower–Middle Devonian succession (Figure 4g). The lowermost splay of
this thrust was most likely penetrated by exploration well 7816/12-1 and consists of phyllitic coal
and sheared coaly shales of the Billefjorden Group (Eide et al., 1991; Figure 4g). Bedding-parallel
thrusts in uppermost Devonian–Mississippian strata are further supported by the presence of an
analogous, sub-horizontal, bedding-parallel fault within the overlying Middle–Upper Triassic
sedimentary rocks of the Barentsøya Formation, which was also penetrated by well 7816/12-1 and
represents a possible early Cenozoic décollement (Eide et al., 1991; see uppermost sub-horizontal
fault in Figure 4g).

Continuous to semi-continuous, parallel, dominantly moderate- to high-amplitude seismic
reflections representing Pennsylvanian–lower Permian sedimentary strata of the Hultberget,
Ebbadalen, Minkinfjellet and Wordiekammen formations thicken eastwards and westwards away
from the fold hinge, i.e., opposite to sedimentary rocks of the Billefjorden Group, and appear to be
affected by much fewer disruptions and, therefore, to be only mildly deformed (Figure 4g).
Pennsylvanian–lower Permian strata are thickest along the eastern fold limb where they are
crosscut by three splays of the early Cenozoic thrust localized along the boundary between the
Billefjorden and Gipsdalen groups and by a steeply east-dipping brittle fault. This steeply east-
dipping fault shows a planar geometry in cross-section, thickening of the Hultberget, Ebbadalen,
Minkinfjellet and Wordiekammen formations in the hanging wall, minor normal offsets of seismic
reflections within these stratigraphic units, and dies out within the lower part of the Wordiekammen
Formation upwards and the upper part of the Lower–Middle Devonian succession downwards.
Based on cross-section geometries, offset kinematics, and thickening of stratigraphic units, this
steeply dipping normal fault is interpreted as a Carboniferous normal fault possibly representing
the southwards continuation of the Billefjorden Fault Zone.

## 5. Discussion





*5.1. Implications of contractional duplexes and décollements in Devonian–Mississippian sedimentary rocks for Ellesmerian and Eurekan deformation*

Uppermost Devonian–Mississippian sedimentary rocks of the Billefjorden Group in Pyramiden (Figure 3b) and Sassenfjorden–Tempelfjorden (Figure 4a, b, d and e) are arranged in gently dipping duplexes comprised of interbedded coal–coaly shale and sandstone deposits with

sigmoidal shear fabrics and (imbricate) link thrusts (McClay and Insley, 1986) connecting bedding-parallel décollements (roof and floor thrusts/detachments; McClay, 1992) localized along lithological boundaries. These structures and geometries are typical in coal deposits reworked by contractional deformation (Phillipson, 2003, 2005; Molinda, 2003; Elizalde et al., 2016). The interpretation of bedding-parallel décollements is supported by minor early Cenozoic thrusts

crosscutting the Hultberget, Ebbadalen, Minkinfjellet and Wordiekammen formations in Sassenfjorden–Tempelfjorden (Figure 1a–b) that flatten downwards and die out within sedimentary strata of the Billefjorden Group (Figure 4c), and by the presence of analogous shallow-dipping, bedding-parallel décollements in uppermost Devonian–Mississippian coals and coaly shales sedimentary strata of the Billefjorden Group in Odellfjellet (Koehl and Muñoz-Barrera,

2018), in Robertsonbreen (between the uppermost Devonian–Mississippian Hørbyebreen Formation and Pennsylvanian–Permian Wordiekammen Formation; Diβmann and Grewing, 1997), in northeastern Bjørnøya (Koehl, in prep.), at Midterhuken and in St. Jonsfjorden (where the unconformity between uppermost Devonian–Mississippian and Pennsylvanian sedimentary rocks possibly acted as a décollement/subhorizontal thrust; Maher and Welbon, 1992; Gasser and

Andresen, 2013; Figure 1a), in Nordenskiöld Land (Braathen and Bergh, 1995), and, potentially, in Oscar II Land (Bergh and Andresen, 1990) and Wedel Jarlsberg Land–Torell Land (Dallmann and Maher, 1989; Figure 1a). Imbrication within the duplexes in Pyramiden indicates top-west thrusting, and most likely reflects Eurekan contraction–transpression since it is the only post-Mississippian episode of contraction–transpression recorded in Spitsbergen. Similar Eurekan

duplex geometries with sigmoidal bedding surfaces and link thrusts were also observed in Triassic strata in Spitsbergen (Andresen et al., 1992; Haremo and Andresen, 1992; Andresen, 2009), thus further supporting an interpretation of early Cenozoic thrusting in Pyramiden.

In Reindalspasset, potential décollements and low-angle thrusts folded into a gentle upright anticline and possibly forming an antiformal thrust stack were identified on seismic data within

Lower–Middle Devonian strata of the Wood Bay and/or Grey Hoek and/or Wijde Bay formations





and uppermost Devonian–Mississippian rocks of the Billefjorden Group (Figure 4g). In tectonically thickened and mildly folded uppermost Devonian–Mississippian rocks, low-angle brittle–ductile thrust faults are comprised of phyllitic (i.e., sheared) and brittle coals (penetrated by well 7816/12-1 at a depth of 2261–2280 meters; Eide et al., 1991) that are similar to sheared

uppermost Devonian–Mississippian coals in Pyramiden, and are arranged into potential duplexes that are comparable to duplexes and thrust systems in uppermost Devonian–Mississippian sedimentary rocks in Pyramiden (Figure 1 and Figure _3_b) and Sassenfjorden–Tempelfjorden (Figure 1, and Figure _4_b and e). The geometries of these duplexes, thrusts and décollements on seismic data in Spitsbergen are similar to analogous structures on seismic data worldwide (e.g.,

Morley et al., 2017, their figure 8). Potential Lower–Middle Devonian rocks show wedge-shaped duplex structures, décollements, folding and thrusting comparable to deformation structures in analogous rocks in Andrée Land, e.g., Bråvallafjella Fold Zone (Piepjohn, 2000; Dallmann and Piepjohn, 2020), and in southern Spitsbergen (e.g., Røkensåta; Figure 1a; Dallmann, 1992), thus potentially supporting the preservation of Devonian sedimentary rocks of the Andrée Land Group

(and/or Mimerdalen Subgroup) east of the Billefjorden Fault Zone in Reindalspasset, pending that the observed normal fault does actually represent the southern continuation of the Billefjorden Fault Zone (Figure 4g; see section 5.3). The presence of décollements within Lower–Middle (–lowermost Upper?) Devonian rocks is further supported by the observation of similar structures between shale and sandstone units of the Wood Bay and Grey Hoek formations in Andrée Land

(Roy, 2007, 2009; Roy et al., unpublished).

        Based on the significant differences in deformation styles, it is probable that the décollements and backward-dipping duplexes in sheared uppermost Devonian–Mississippian coals–coaly shales decoupled early Cenozoic Eurekan deformation between folded, shale-rich, Lower Devonian rocks and undeformed to poorly-deformed uppermost Devonian–Permian

sedimentary strata in Pyramiden (Figure 2 and Figure _3_b). Seismic data in Sassenfjorden–Tempelfjorden also show potential duplexes and décollements within uppermost Devonian–Mississippian coal-rich deposits (Figure 4a, b, d and e). In these fjords, steeply dipping, basement-seated brittle faults seem to have propagated upwards during early Cenozoic Eurekan deformation, resulting in fault-propagation folding and reverse offsets in uppermost Devonian–Permian

sedimentary strata (Figure 4a and c). These faults die out upwards within uppermost Devonian–Pennsylvanian sedimentary rocks, while minor early Cenozoic thrusts crosscutting Pennsylvanian–



Permian sedimentary strata appear to flatten downwards and die out into high-amplitude seismic reflections interpreted as uppermost Devonian–Mississippian coals, thus, also suggesting decoupling of Eurekan deformation by early Cenozoic décollements in uppermost Devonian–

Mississippian coals of the Billefjorden Group.

In Reindalspasset, early Cenozoic duplexes and thrusts within potential Lower–Middle Devonian strata of the Wood Bay and/or Grey Hoek and/or Widje Bay formations die out upwards and minor thrusts within Pennsylvanian–Permian rocks die out downwards near or at the boundary with coal-rich sedimentary rocks of the Billefjorden Group (Figure 4g), thus also supporting the

presence of early Cenozoic décollements within uppermost Devonian–Mississippian coaly shales and coals and (partial) decoupling of Eurekan deformation. Thickened coal-rich deposits are long known to be able to decouple deformation both in contractional (Frodsham and Gayer, 1999, their figures 1b, 2, 7 and 9) and extensional settings (Wilson and Wojtal, 1986, their figures 7 and 10). In Svalbard, recent field studies by Koehl and Muñoz-Barrera (2018) in the northern part of the

Billefjorden Trough in Odellfjellet (Figure 1b) showed that bedding-parallel duplex-shaped décollements in uppermost Devonian–Mississippian coaly shales may have partly inhibited early Cenozoic Eurekan contraction–transpression in overlying Pennsylvanian strata, thus further supporting the presence of such décollements in Pyramiden (Figure 3b), Sassenfjorden– Tempelfjorden (Figure 4a–f) and Reindalspasset (Figure 4g).

Uppermost Devonian–Mississippian coal-rich strata are locally thicker in Pyramiden, thus resulting in their exploitation by Russia until the early 90s (Livshitz, 1966; Cutbill et al., 1976). They are also thicker in Sassenfjorden in the hanging wall of the east-dipping Billefjorden Fault Zone near the intersection with a NNE-dipping basement-seated fault (Figure 1, and Figure _4_a–d), and within the hinge zone of the anticline adjacent to the possible southward continuation of the

Billefjorden Fault Zone in Reindalspasset (Figure 4g). Recent studies of sedimentary rocks of the Billefjorden Group in the Ottar Basin (Tonstad, 2018), the Finnmark Platform (Koehl et al., 2018) in the SW Barents Sea, and the northern part of the Billefjorden Trough (Koehl and Muñoz-Barrera, 2018) show that uppermost Devonian–Mississippian sedimentary strata were deposited into subsiding basins bounded by normal faults. In addition, high-amplitude seismic reflections in the

Ottar Basin representing thickened, coal-rich, uppermost Devonian–Mississippian sedimentary strata analog to those observed in Sassenfjorden–Tempelfjorden are thickest on basin edges where fluvial systems dominated in latest Devonian–Mississippian times (Tonstad, 2018). It is possible





that, in Spitsbergen too, thick uppermost Devonian–Mississippian coal seams are restricted to the basin edges along boundary faults, thus explaining the localization of contractional duplexes and

décollements in areas such as Pyramiden, Sassenfjorden, Reindalspasset and (potentially) Triungen during early Cenozoic deformation, partially decoupling deformation between Lower–lowermost Upper Devonian sedimentary rocks of the Andrée Land Group and Mimerdalen Subgroup and thick Pennsylvanian–Permian deposits of the Gipsdalen Group, and locally shielding the latter from Eurekan deformation, while Pennsylvanian sedimentary rocks in basinal areas in the hanging wall

of the Odellfjellet Fault were involved in Eurekan deformation, and Carboniferous normal faults were inverted, e.g., in Odellfjellet (Koehl and Muñoz-Barrera, 2018), Løvehovden–Hultberget (Dallmann, 1993; Maher and Braathen, 2011), Adolfbukta (Harland et al., 1988), Lykteneset (Koehl et al., submitted), Anservika (Ringset and Andresen, 1988), and Sassenfjorden (Figure 4a–f).

Based on field and seismic data in central Spitsbergen (present study; Koehl and Muñoz-Barrera, 2018; Koehl et al., submitted) and on analog modelling (Bonini, 2001), it is possible that Lower–lowermost Upper Devonian sedimentary deposits of the Andrée Land Group and Mimerdalen Subgroup were folded exclusively in early Cenozoic times since the differences in deformation style and intensity between Devonian and Carboniferous–Permian deposits can be

explained simply by decoupling of Eurekan deformation by weak, uppermost Devonian–Mississippian, coal- and shale-rich sedimentary deposits of the Billefjorden Group (Figs. Figure 3b, and Figure *4*a–e and g; Koehl and Muñoz-Barrera, 2018). Hence, a short-lived episode of Late Devonian (Ellesmerian) deformation is not required to explain differential deformation within Lower Devonian to Permian sedimentary successions in central Spitsbergen, thus potentially

simplifying the late Paleozoic tectonic history of the area by reducing it to the Caledonian Orogeny and late–post-Caledonian extensional collapse–rifting. This is further supported by a field study in Robertsonbreen (central Spitsbergen; Figure 1b), where Diβmann and Grewing (1997) noticed that sedimentary strata of the lowermost Upper Devonian Plantekløfta Formation and uppermost Devonian–Mississippian Hørbyebreen Formation are both similarly folded, i.e., suggesting that

early Cenozoic deformation may be (at least partially) responsible for folding of Lower–lowermost Upper Devonian rocks of the Andrée Land Group and Mimerdalen Subgroup in central Spitsbergen.





Strain decoupling, décollements and contractional duplexes are common features in the West Spitsbergen Fold-and-Thrust Belt and were described at various locations and within varied rock types and stratigraphic units. Notably, Ringset and Andresen (1988) and Harland et al. (1988) discussed the presence of subhorizontal, bedding-parallel décollements within Pennsylvanian evaporites of the Ebbadalen and Minkinfjellet formations in eastern Billefjorden, from which early Cenozoic Eurekan thrusts may have ramped upwards into trailing imbricate fans (Boyer and Elliott, 1982) due to lateral lithological variations within Pennsylvanian formations (Ringset and Andresen, 1988). In addition, in western Spitsbergen, Maher (1988), Saalmann and Thiedig (2000) and Bergh and Andresen (1990) described early Cenozoic décollements and gently hinterland-dipping duplexes in uppermost Pennsylvanian–Permian sedimentary deposits of the Wordiekammen, Gipshuken and Kapp Starostin formations, which may represent analogs to duplex structures and associated bedding-parallel décollements and low-angle thrusts within uppermost Devonian–Mississippian coals and coaly shales in Pyramiden, Sassenfjorden–Tempelfjorden and Reindalspasset (Figure 3b and Figure *4*). Noteworthy, a model of critical wedge taper for the West Spitsbergen Fold-and-Thrust Belt predicted an increasing influence of decoupling (as observed in Pyramiden, Sassenfjorden–Tempelfjorden and Reindalspasset; Figure 3b and Figure *4*) towards the foreland of the fold and thrust belt, i.e., near the study area in central Spitsbergen (Braathen et al., 1999).

### 5.2. Formation mechanism for duplexes and décollements in uppermost Devonian–Mississippian rocks in Pyramiden

Backward-dipping duplexes in Pyramiden are juxtaposed against east-dipping (and locally overturned west-dipping) Devonian strata of the Andrée Land Group and Mimerdalen Subgroup (Figures 2 and 3a and b) adjacent to and showing similar attitude to major fold structures in Mimerdalen thus far ascribed to the Ellesmerian Orogeny (Vogt, 1938; Piepjohn, 2000; Bergh et al., 2011). It is possible that, during early Cenozoic folding, Lower–lowermost Upper Devonian rocks of the Andrée Land Group and Mimerdalen Subgroup in the west may have acted as a relatively rigid buttress, i.e., partly deforming but not as easily as overlying weak uppermost Devonian–Mississippian coals and coaly shales of the Billefjorden Group that localized the formation of duplexes and décollements, and, thus, allowing these structures to ramp upwards to the west. This is supported by field studies (Fard et al., 2006) and analog modelling (Bahroudi and





Koyi, 2003) in the Zagros Fold-and-Thrust Belt showing buttressing, backward-dipping duplexes

and décollements in the hanging wall of deep-seated faults, and by analog modelling of décollements in weak sedimentary layers with limited lateral extent (Costa and Vendeville, 2002, their model 3). Notably, Costa and Vendeville's model shows that initially sub-horizontal sedimentary strata may have been tilted backwards (i.e., eastwards in Pyramiden) during contraction, and that décollement lithology (i.e., uppermost Devonian–Mississippian coal–coaly

shale) may be incorporated and transported (top-west to top-WNW in Pyramiden; Figure 3a) as part of the hanging wall sequence during thrusting. In Pyramiden, this is supported by drill data from Trust Arktikugol showing that coal seams of the Billefjorden Group at the mine continue eastwards and preserve a gentle–moderate dip to the east (Aakvik, 1981, his figure 8.2.5). This interpretation implies the presence of the Balliolbreen Fault in Pyramiden, which is discussed in

section 5.3.

Another possibility is that the Pyramiden outcrop represents a mildly inverted extensional fault-block that was gently folded due to upward propagation of the Balliolbreen Fault (if present at all in Pyramiden; see section 5.3) and Odellfjellet Fault (e.g., gentle tilt to the east-southeast of strata of the Minkinfjellet Formation in Pyramiden; Koehl et al., 2016). Fault-propagation folds

(Schlische, 1995) were discussed along the Løvehovden Fault (Maher and Braathen, 2011) and Billefjorden Fault Zone (Braathen et al., 2011; Bælum and Braathen, 2012) in cental Spitsbergen. However, this model implies the existence of the Balliolbreen Fault in Pyramiden as a steeply east-dipping fault, which is not obvious (see section 5.3), and, alone, does not explain the presence of bedding-parallel décollements and backward-dipping duplexes within uppermost Devonian–

Mississippian coals and coaly shales of the Billefjorden Group in Pyramiden and Sassenfjorden–Tempelfjorden (Figure 3b, and Figure _4_b and e). Moreover, seismic data in Reindalspasset show that a steeply east-dipping normal fault potentially representing the southwards continuation of the Billefjorden Fault Zone (Odellfjellet Fault?) is located along the eastern flank of a broad, gentle anticline (Figure 4g) and, hence, might be related to (or might have interacted with) the fold

structure but is most likely not the cause of folding in this area.

Analog modelling of inversion in asymmetric half-graben basins shows features similar to those observed in Pyramiden, demonstrating a potential relationship between weak, early syn-rift sedimentary deposits and segments of basin-bounding faults (Buiter and Pfiffner, 2003, their figure 6a). Notably, in presence of weak, syn-rift sedimentary rocks in basin-edge fault-blocks, newly-



formed shortcut shear zones or faults (McClay, 1989) may branch off preexisting inverted basin-
       bounding normal faults, and ramp up into the weak, syn-rift sedimentary strata, potentially using
       décollement levels to accommodate contraction. Buiter and Pfiffner (2003) further argue that
       basement blocks experience much less contraction-related rotation along preexisting normal faults.
       Thus, a possible scenario for the early Cenozoic tectonic history of the Billefjorden Fault Zone in
Pyramiden might involve the formation of a shortcut shear zone or fault along an inverted portion
       of the Billefjorden Fault Zone at depth, branching off and ramping upwards into, weak, coal- and
       coaly shale-dominated syn-rift sedimentary rocks of the Billefjorden Group, forming bedding-
       parallel décollements (Phillipson, 2003, 2005; Molinda, 2003; Elizalde et al., 2016) and east-
       dipping, backward-dipping duplexes (Figure 3b).

Alternatively, early Cenozoic reverse reactivation/overprinting of the potentially upward-
       flattening Balliolbreen Fault (if present at all in Pyramiden; see section 5.3) might have triggered
       the development of a décollement within and of a fault-bend hanging wall anticline (e.g., the Kuqa
       Fold Belt in northwestern China; Wang et al., 2013; Izquierdo-Llavall et al., 2017) above
       uppermost Devonian–Mississippian coals, e.g., in Reindalspasset (Figure 4g). In this scenario,
backward-dipping duplexes and décollements in uppermost Devonian–Mississippian coals–coaly
       shales may have acted as a roof décollement decoupling uppermost Devonian–Permian strata from
       (Lower–lowermost Upper) Devonian rocks, passively thrusting the former over the latter (Bonini,
       2001). Through this process, the length of the roof sequence (uppermost Devonian–Permian
       sedimentary strata) remains essentially the same, whereas the length of the floor sequence (Lower–
lowermost Upper Devonian rocks of the Andrée Land Group and Mimerdalen Subgroup) decreases
       through intense folding (Bonini, 2001). This may (partially) explain the significant differences of
       deformation between folded Lower–lowermost Upper Devonian of the Andrée Land
       Group/Mimerdalen Subgroup (Vogt, 1938; Harland et al., 1974; Piepjohn et al., 1997; Michaelsen
       et al., 1997; Michaelsen, 1998; Piepjohn, 2000), strongly sheared uppermost Devonian–
Mississippian strata of the Billefjorden Group (Figure 3b), and poorly deformed to gently tilted
       uppermost Devonian–Permian strata of the Billefjorden and Gipsdalen groups in central
       Spitsbergen (e.g., Braathen et al., 2011) without a short-lived episode of Ellesmerian contraction
       in the Late Devonian. The lack of uppermost Devonian–Mississippian coals and coaly shales of
       the Billefjorden Group directly on top of folded Lower (–lowermost Upper?) Devonian
sedimentary rocks above the mine entrance in Pyramiden may suggest that uppermost Devonian–





Mississippian coals–coaly shales were too thin or too localized (syn-rift?) to allow décollements to ramp all the way up to the mine entrance or that early Cenozoic Eurekan contraction–transpression was too mild to form a complete ramp-anticline (assuming that the Balliolbreen Fault is present in Pyramiden) with roof décollement over Lower Devonian sedimentary rocks (e.g., Faisal and Dixon,
2015).

Another plausible interpretation might be that of (a) west-directed imbricate fan(s) in Pennsylvanian evaporitic deposits and/or uppermost Devonian–Mississippian coals and coaly shales at depth in the Billefjorden Trough with east-dipping imbricate thrusts ramping upwards into coals and coaly shales of the Billefjorden Group in the footwall of the Odellfjellet Fault, in
Pyramiden. This interpretation is supported by field studies of Ringset and Andresen (1988) who discussed imbricate (thrust) fans and associated basal décollement developed along lithological boundaries within the Ebbadalen Formation in Anservika–Gipshuken (see Figure 1b for location), Harland et al. (1988) who described sheared evaporites within the Ebbadalen and Gipshuken formations in eastern Billefjorden, and by recent field studies showing the presence of a potentially
gently east-dipping, bedding-parallel thrust–décollement within the Billefjorden Group and Hultberget Formation in Anservika (Henningsen et al., pers. comm. 2019), and within the Hultberget Formation in Lykteneset (Koehl et al., submitted).

Based on field data, backward-dipping duplexes and bedding-parallel décollements in uppermost Devonian–Mississippian coals and coaly shales of the Billefjorden Group in Pyramiden
are believed to have formed through a combination of at least two or more mechanisms, including Devonian rocks of the Andrée Land Group and Mimerdalen Subgroup acting as a relatively rigid buttress to the west (e.g., Figure 4g), fault-propagation folding of (a) preexisting fault(s) like the Balliolbreen Fault and/or Odellfjellet Fault (although not very likely), shortcut faulting propagating upwards and westwards from the Billefjorden Fault Zone (e.g., Buiter and Pfiffner, 2003),
ramp/fault-bend hanging wall anticline with roof décollement (e.g., Faisal and Dixon, 2015), and imbricate fan with basal décollement in the Billefjorden Trough (e.g., Ringset and Andresen, 1988; Henningsen et al., pers. comm. 2019).

*5.3. Geometry and kinematics of the Balliolbreen Fault and implications for Ellesmerian and*
*Eurekan deformation events, and Carboniferous normal faulting*





Structural field analysis in the gully below the entrance of the Russian coal mine in Pyramiden has shown the presence of a sub-vertical, steeply east-dipping brittle fault tentatively interpreted as the Balliolbreen Fault and comprised of cataclastic fault rock that, half-way to the mine, crosscuts steeply east-dipping, quartzitic, (Lower–lowermost Upper?) Devonian sedimentary rocks involved in a fold structure with bedding locally overturned to the east (Figure 2 and Figure _3_a, and supplement 3). Thin section analysis on both sides of this fault (supplement 1) shows cataclased (Lower–lowermost Upper?) Devonian sandstone both in the fault footwall and hanging wall, suggesting that there are no basement rocks at this locality, which is supported by geological maps of Harland et al. (1974), Aakvik (1981), Lamar et al. (1986), and geological maps and logs of Trust Arktikugol (1988; Sirotkin, pers. comm. 2019). In addition, the steeply east-dipping fault does not seem to extend upwards into overlying uppermost Devonian–Mississippian clastic deposits above phyllitic coal-rich sedimentary strata. It is possible that the décollements within uppermost Devonian–Mississippian coals–coaly shales represent the upward low-angle continuation of the steeply east-dipping fault, but the structural location of the décollements (almost directly over the fault) would require an abrupt change of geometry of the fault from subvertical to low-angle (c. 30°; Figure 3b) within a narrow zone, which is unlikely. In addition, fault surfaces and lithological transitions switch from dominant N–S to NNW–SSE strikes and trends in uppermost Devonian–Mississippian coals–coaly shales below the coal-mine entrance (Figure 2 and Figure _3_a, and stereonet 3 in Figure 2) to dominantly WNW–ESE in Lower (–lowermost Upper?) Devonian rocks and uppermost Devonian–Mississippian sandstone above the mine entrance (Figure 2 and Figure _3_a, and stereonet 2 in Figure 2), i.e., parallel to most outcrops sections of uppermost Devonian–Mississippian strata in this part of the study area.

Above the coal mine in Pyramiden, the contact between Lower (–lowermost Upper?) Devonian sedimentary strata and uppermost Devonian–Mississippian sedimentary rocks is not clearly exposed (partly loose blocks) and its nature is relatively speculative. It may be (1) a (folded?) stratigraphic unconformity and/or (2) a bedding-parallel décollement. Based on the internal geometry of bedding surfaces and deformation state of uppermost Devonian–Mississippian sedimentary strata of the Billefjorden Group, which are arranged into contractional, west-verging duplexes separated by low-angle, bedding-parallel décollements (Figure 3b), it is possible that the stratigraphic contact hosts a décollement, e.g., the potential prolongation of one of the décollements within coal- and coaly shale-rich deposits of the Billefjorden Group (Figure 2 and Figure _3_b).





However, uppermost Devonian–Mississippian deposits above the coal mine appear to consist only of clastic deposits and, hence, lack weak coals–coaly shales into which décollements preferentially localize. Thus, the contact between Lower (–lowermost Upper?) Devonian and uppermost Devonian–Mississippian sedimentary rocks above the mine in Pyramiden most likely corresponds to a (folded?) unconformity.

Even if the décollements within uppermost Devonian–Mississippian coals and shales (Figure 3b) were to represent the upwards continuation of the steeply east-dipping fault (Figure 2), these most likely do not extend into Lower (–lowermost Upper?) Devonian and uppermost Devonian–Mississippian sandstone units above the mine entrance. Based on the similarity between the strike and dip of the steeply east-dipping fault and the trend and dip of (locally overturned) Lower Devonian bedding surfaces in Pyramiden (Figure 2 and Figure *3*a), it is possible that the steeply east-dipping fault formed as a minor, bedding-parallel (fold-limb parallel) fault related to post-Caledonian gravitional collapse processes and low-angle detachments (e.g., the Woodfjorden detachment in Andrée Land; Roy, 2007, 2009; Roy et al., unpublished; Figure 1a) in Lower–lowermost Upper Devonian sedimentary rocks of the Andrée Land Group and Mimerdalen Subgroup in northern Spitsbergen (e.g., Chorowicz, 1992), or formed as a minor, bedding-parallel Eurekan accommodation thrust (e.g., Cosgrove, 2015) in the early Cenozoic. Since no major fault was identified in Pyramiden it is probable that the Balliolbreen Fault does not crop out or is not present there. This is supported by microsctructures along the steeply east-dipping fault in Pyramiden (Figure 2), e.g., mild undulose extinction and limited recrystallization and low amounts of displacement along distributed brittle cracks in fault rock (supplement 1), which indicate mild deformation associated with low-grade pressure–temperature conditions (< 280°C; Stipp et al. 2002).

In Reindalspasset, the planar, east-dipping normal fault that offsets Pennsylvanian–lower Permian sedimentary rocks may represent the potential continuation of the basin-bounding Odellfjellet Fault ("BFZ?" in Figure 4g), and the Eurekan thrust (and associated splays) localized along the boundary between uppermost Devonian–Mississippian and Pennsylvanian sedimentary successions (Figure 4g) the continuation of the (inverted?) Balliolbreen Fault. Fault relationships in cross section in Reindalspasset are comparable to what is proposed for the Balliolbreen and Odellfjellet faults in Pyramiden, e.g., possible merging at depth and hundreds of meter- to kilometer-scale lateral spacing between the faults (see previous section), assuming that the



Balliolbreen Fault is present in Pyramiden. The preservation of Lower–lowermost Upper Devonian
sedimentary rocks of the Andrée Land Group and/or Mimerdalen Subgroup east of the Billefjorden

Fault Zone in Reindalspasset suggests that this fault did not accommodate top-west reverse
movement in Late Devonian times as proposed by previous works in Dickson Land (Vogt, 1938;
Friend, 1961; Piepjohn, 2000; Dallmann and Piepjohn, 2020). Would such movements have
occurred, Devonian sedimentary rocks of the Andrée Land Group and/or Mimerdalen Subgroup in
the upthrusted block east of the Billefjorden Fault Zone would have been exposed and subjected to

continental erosion. This is clearly not the case in Reindalspasset where potential Lower–
lowermost Upper Devonian sedimentary rock units appear to thicken eastwards. The presence of
Devonian sedimentary east of the Billefjorden Fault Zone in Reindalspasset is also supported by
the interpretation of well bore data in the Raddedalen-1 well in Edgeøya (Harland and Kelly, 1997).

Based on field data in Pyramiden and seismic data in Sassenfjorden and Reindalspasset,

and on previous work (Harland et al., 1974; Lamar et al., 1982, 1986; McCann, 1993; Lamar and
Douglass, 1995), the Balliolbreen Fault displays significant along-strike variations in geometry and
kinematics. In the north, in Odellfjellet and Sentinelfjellet (Figure 1b), the Balliolbreen Fault dips
c. 60–65° to the east and juxtaposes Precambrian basement unconformably overlain by uppermost
Devonian–Mississippian strata of the Billefjorden Group in the hanging wall against Lower

Devonian strata of the Wood Bay Formation supposedly unconformably overlain by uppermost
Devonian–Mississippian rocks of the Billefjorden Group (Harland et al., 1974; Lamar et al., 1986;
Lamar and Douglass, 1995). Both in Odellfjellet and Sentinelfjellet, it is unclear whether the
Balliolbreen Fault offsets uppermost Devonian–Mississippian strata, or if the fault is
unconformably overlain by uppermost Devonian–Mississippian rocks (Lamar et al., 1982, 1986;

Lamar and Douglass, 1995). Although Harland et al. (1974) argue that the Triungen Member of
the Hørbyebreen Formation is unfaulted in Sentinelfjellet (thus potentially supporting Late
Devonian top-west thrusting along the Balliolbreen Fault and no further reactivation), stratigraphic
contacts in this area are covered by screes and poorly–not exposed (like in Triungen in Figure 3c–
e) and inaccessible because located on very steep slopes–cliffs (see toposvalbard.npolar.no). The

presence of newly evidenced décollements in the lower part of the Billefjorden Group in
Pyramiden, Sassenfjorden–Tempelfjorden and Reindalspasset (Figure 3b and Figure _4_) suggest
that the nature of the contact of the Billefjorden Group with underlying Devonian rock units must
be interpreted with care, especially were covered by screes. If sedimentary strata of the Billefjorden





Group are actually truncated by the Balliolbreen Fault in Sentinelfjellet (e.g., McCann, 1993, his
figures 5.9 and 5.10), then early Cenozoic thrusting may, in conjunction with Carboniferous normal
faulting, explain the observed juxtaposition of Precambrian basement and Lower Devonian
sedimentary rocks (Figure 5). In this scenario, basement rocks constituting the Caledonian
Atomfjella Antiform were located close to the surface at the end of the Caledonian Orogeny, thus
leaving no (or limited) accommodation space east of the Billefjorden Fault Zone in Ny-Friesland
(Figure 1a) during Devonian sedimentation sourced from the collapsing orogen and exhuming core
complexes (e.g., Bockfjorden Anticline; Braathen et al., 2018; Figure 5a). In the Carboniferous,
normal faulting and footwall rotation along the Odellfjellet Fault possibly exhumed a small portion
of basement rocks in the footwall of the fault (Figure 5b–c), and subsequent early Cenozoic
deformation may have thrusted part of the exposed basement rocks in the footwall as a kilometer-
scale lens along a possibly inverted Carboniferous normal fault, the Balliolbreen Fault (Figure 5d)
potentially leading to extensive deformation of Lower–lowermost Upper Devonian rocks in
Dickson Land, which acted as a buttress absorbing most of Eurekan deformation together with
sheared uppermost Devonian–Mississippian coals and shales of the Billefjorden Group (note that
deformation within Lower–lowermost Upper Devonian rocks is not detailed in Figure 5d). In this
model, Carboniferous normal and early Cenozoic reverse offsets along the Balliolbreen Fault have
similar magnitude, as shown in Mumien (juxtaposition of the Ebbadalen Formation and
Billefjorden Group against the Wordiekammen Formation and the Billefjorden Group with no
apparent offset at top Billefjorden Group level; Dallmann et al., 2004; Dallmann, 2015; Figure 1b),
and in Sentinelfjellet and Odellfjellet (top of Billefjorden Group offset by 0–40 m; Harland et al.,
1974; Lamar et al., 1986; Figure 5e). Thus, it is possible that the above mentioned localities reflect
different structural levels of the same fault system (Figure 5e). Eurekan inversion of Carboniferous
normal faults in central Spitsbergen is also supported by reverse offset and thrust-related folding
along the Overgangshytta fault in Odellfjellet (Koehl and Muñoz-Barrera, 2018), and by minor
reverse offset of thickened, uppermost Devonian–Mississippian and Pennsylvanian sedimentary
deposits in the hanging wall of the east-dipping Billefjorden Fault Zone, near the intersection with
a steeply NNE-dipping basement-seated fault in Sassenfjorden (Figure 4b).

The high degree of uncertainty in the relationship (truncated or truncating) between the
Balliolbreen Fault and uppermost Devonian–Mississippian sedimentary strata of the Billefjorden
Group (especially in Odellfjellet and Sentinelfjellet; Harland et al., 1974; Lamar et al., 1986; Lamar



and Douglass, 1995), and the uncertainty regarding the nature of the contact (unconformity or
       bedding-parallel décollements–thrusts) between Lower Devonian and uppermost Devonian–
       Mississippian sedimentary strata shed by the presence of bedding-parallel Eurekan décollements
       and thrusts in Pyramiden (Figure 3b), Sassenfjorden–Tempelfjorden (Figure 4b, c and e) and
       Reindalspasset (Figure 4g) call for caution and further (re-) examination of outcrops of uppermost

Devonian–Mississippian rocks along the Balliolbreen Fault in central Spitsbergen. Notably, the
       significant along strike differences in cross-section geometry from subvertical, e.g., in Pyramiden
       (if present at all; Figure 2 and Figure _3_b) to shallow dipping, e.g., in Reindalspasset (Eurekan thrust
       localized along the Billefjorden–Gipsdalen groups boundary; Figure 4g), together with the strong
       contrasts in offset stratigraphic units, e.g., Pennsylvanian rocks of the Ebbadalen Formation

overlain by carbonates of the Wordiekammen Formation in the hanging wall against Lower
       Devonian rocks of the Wood Bay Formation unconformably overlain by strata of the
       Wordiekammen Formation in the footwall in Yggdrasilkampen (Dallmann et al., 2004; Figure 1b),
       Pennsylvanian Ebbadalen Formation against uppermost Pennsylvanian–lower Permian
       Wordiekammen Formation in Mumien (Dallmann et al., 2004; Dallmann, 2015), Lower Devonian

rocks overlain by uppermost Devonian–Mississippian Billefjorden Group in the hanging wall
       against Lower Devonian rocks in the footwall in Pyramiden (if present at all; Figure 2 and Figure
       _3_a), Precambrian basement rocks in the hanging wall against Lower Devonian rocks in the footwall
       in Odellfjellet and Sentinelfjellet (Harland et al., 1974; Lamar et al., 1986), and in inferred timing
       and kinematics, e.g., Carboniferous normal faulting in Yggdrasilkampen (Dallmann et al., 2004),

early Cenozoic reverse movement in Pyramiden (if present at all; Figure 3b) and possibly in
       Reindalspasset (if present at all; Figure 4g) and Flowerdalen (Harland et al., 1974; Haremo et al.,
       1990; Haremo and Andresen, 1992), Carboniferous normal and early Cenozoic reverse faulting in
       Mumien (Dallmann, 2015), and potential Late Devonian (e.g., Harland et al., 1974; Piepjohn, 2000;
       Dallmann and Piepjohn, 2020) or early Cenozoic thrusting (this study; Koehl and Muñoz-Barrera,

2018) in Odellfjellet and Sentinelfjellet, suggest that the Balliolbreen Fault might consist of several,
       discrete, disconnected (soft-linked and/or stepping?) or possibly offset fault segments crosscut by
       suborthogonal faults (McCann, 1993, his figure 5.11; Koehl, 2020). For example, a basement-
       seated reverse fault in Sassenfjorden–Tempelfjorden accommodated top-SSW thrusting during
       Eurekan deformation (Figure 4a–b) and seem to have limited the amount of Eurekan

reactivation/overprinting (strain partitioning) along east-dipping segments of the Billefjorden Fault





Zone in this area, which shows mainly down-east Carboniferous normal offset with limited amount of early Cenozoic reworking along the main east-dipping fault (e.g., Figure 4d), and may be responsible for restricting sediment deposition/preservation to the southwest of Sassenfjorden during Eurekan tectonism in the early Cenozoic, thus, explaining sediment province from the

northeast (e.g., Petersen et al., 2016). Another example where strain partitioning may have occurred along suborthogonal faults is Yggdrasilkampen, where the possible continuation of the Balliolbreen Fault juxtaposes Pennsylvanian (hanging wall) against Lower Devonian (footwall) sedimentary rocks suggesting that Carboniferous normal faulting was followed by limited early Cenozoic reactivation/overprinting if any at all. The character of the Billefjorden Fault Zone in

Sassenfjorden and Yggdrasilkampen contrasts sharply with areas farther north (e.g., in Sentinelfjellet and Odellfjellet; Harland et al., 1974; Lamar et al., 1986; Lamar and Douglass, 1995; Dallmann et al., 2004; Dallmann, 2015) and farther south (e.g., in Flowerdalen; Harland et al., 1974; Haremo et al., 1990; Haremo and Andresen, 1992; Figure 1b) where the east-dipping Billefjorden Fault Zone displays clear evidence of top-west Eurekan movements. More of these

(WNW–ESE-striking) suborthogonal faults are described and their impact on Eurekan strain partitioning further discussed in Koehl et al. (submitted).

## 6. Conclusion

1. Thickened uppermost Devonian–Mississippian sedimentary deposits of the Billefjorden Group in central Spitsbergen are arranged in duplexes comprised of phyllitic coal–coaly shale interbedded with sandstone showing sigmoidal shear fabrics separated by imbricate thrusts linking an upper (roof thrust) and a lower (floor thrust) décollements that localized along lithological transitions.

2. Early Cenozoic bedding-parallel décollements and thrusts in tectonically thickened, coal-rich sedimentary rocks of the Billefjorden Group, in the Wordiekammen Formation, and in Lower–lowermost Upper Devonian sedimentary rocks partially decoupled Eurekan deformation, resulting in intense folding in Devonian sedimentary rocks of the Andrée Land Group and Mimerdalen Subgroup and uppermost Devonian–Mississippian coals of the

Billefjorden Group, and mild to no deformation in Carboniferous–Permian strata in central Spitsbergen, thus suggesting that Late Devonian Ellesmerian contraction is not required to





explain differential deformation within Lower Devonian to Permian sedimentary successions in central Spitsbergen.

3. Early Cenozoic backward-dipping duplexes and bedding-parallel décollements in the Billefjorden Group in Pyramiden formed through shortcut faulting propagating upwards and westwards from the Odellfjellet Fault, and/or as roof décollements of a ramp/fault-bend hanging wall anticline, and/or as part of an imbricate fan with basal décollement in the Billefjorden Trough. Early Cenozoic contractional structures in uppermost Devonian–Mississippian coals–coaly shales also include fault-propagation folds over preexisting basement-seated faults in Sassenfjorden, and a possible antiformal thrust stack (or ramp anticline) in Reindalspasset.

4. Lower–lowermost Upper Devonian sedimentary rocks might be preserved east of the Billefjorden Fault Zone in Reindalspasset, thus suggesting that the Billefjorden Fault Zone did not act as a reverse fault in Late Devonian times.

5. Thrusting of Proterozoic basement rocks over Lower Devonian sedimentary rocks along the Balliolbreen Fault and fold structures within strata of the Andrée Land Group and Mimerdalen Subgroup in central Spitsbergen may be explained by a combination of down-east Carboniferous normal faulting with associated footwall rotation and exhumation, and subsequent top-west early Cenozoic Eurekan thrusting along the Billefjorden Fault Zone

6. The uncertain relationship of the Balliolbreen Fault with uppermost Devonian–Mississippian sedimentary strata and the poorly constrained nature of the contact (unconformity or bedding-parallel décollements–thrusts) between Devonian of the Andrée Land Group and Mimerdalen Subgroup and uppermost Devonian–Mississippian sedimentary strata, as well as significant along strike variations in cross-section geometry, offset stratigraphy, and inferred timing and kinematics suggest that the Balliolbreen Fault consists of several, discrete, unconnected (soft-linked and/or stepping) or most likely offset fault segments that were reactivated/overprinted with varying degree during Eurekan deformation due to strain partitioning along suborthogonal Eurekan reverse faults.

**Data availability**





High-resolution versions of the figures and supplements of the manuscript necessary to identify individual reflections and structures can be found at DataverseNO (https://doi.org/10.18710/MXKQPE).

**Competing interests**

The author declares that he has no conflict of interest.

**Acknowledgements**

The present study is part of the ARCEx (Research Centre for Arctic Petroleum Exploration), which is funded by the Research Council of Norway (grant number 228107) together
with 10 academic and eight industry partners, and of the SEAMSTRESS project supported by a starting grant of the Tromsø Research Foundation and the Research Council of Norway (grant number 287865). The author would like to thank all the persons from these institutions that are involved in this project. Most grateful thanks to Prof. John Marshall (University of Southampton),
Dr. Christopher Berry (University of Cardiff), Dr. Gilda Lopes (University of Algarve), and Prof. Gunn Mangerud (University of Bergen) for their help with the palynology of Devonian–Mississippian rocks in Spitsbergen. Many thanks to Alexander Andreas for field collaborations, to Prof. Steffen Bergh, Dr. Winfried Dallmann, Prof. Holger Stunitz and Dr. Mélanie Forien (University of Tromsø), and Dr. Karsten Piepjohn (BGR) and Assoc. Prof. Jaroslaw Majka
(Uppsala University) for fruitful discussions. Sebastian Sikora and the University Centre in Svalbard (UNIS) are thanked for boat-transportation to Pyramiden in summer 2016. The Norwegian Petroleum Directorate, Equinor and Store Norske Spitsbergen Kulkompani are thanked for granting access to seismic and well data. Many thanks to Ivar Stokkeland from the Norwegian Polar Institute Library in Tromsø, Norway, for his help with finding old, non-digitized publications
about the geology of Svalbard (available at the Norwegian Polar Institute Library in Tromsø, Norway; list of publications included in Appendix A). The Ph.D. Thesis of John G. Gjelberg (1984) was also digitized and, thanks to the University of Bergen and to John G. Gjelberg's family, is now available from the University of Bergen Library at http://bora.uib.no/handle/1956/20981.

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







**Figure 1:** **(a) Topographic–bathymetric map around Spitsbergen modified after Jakobsson et al. (2012). Abbreviations: Bi: Billefjorden; Bk: Bockfjorden; Kg: Kongsfjorden; Mi: Midterhuken; Ra: Raudfjorden; Re: Reindalspasset; Rø: Røkensåta; Ss: Sassenfjorden; SJ: St-Jonsfjorden; Tp: Tempelfjorden; Tr: Triungen; (b) Geological map modified from svalbardkartet.npolar.no showing the main tectono-stratigraphic units and structures in the study area in central Spitsbergen. Abbreviations: AA: Atomfjella Antiform; Af: Adolfbukta; An: Anservika; BF: Balliolbreen Fault; Fw: Flowerdalen; Gh: Gipshuken; Ly: Lykteneset; Lø: Løvehovden–Hultberget; Mu: Mumien; Od: Odellfjellet; OF: Odelfjellet Fault; Py: Pyramiden; Re: Reindalspasset; Rs: Robertsonbreen; RT: Robertsonbreen thrust; Se: Sentinelfjellet; TGFZ: Triungen–Grønhorgdalen Fault Zone; Tr: Triungen; Yg: Yggdrasilkampen.**





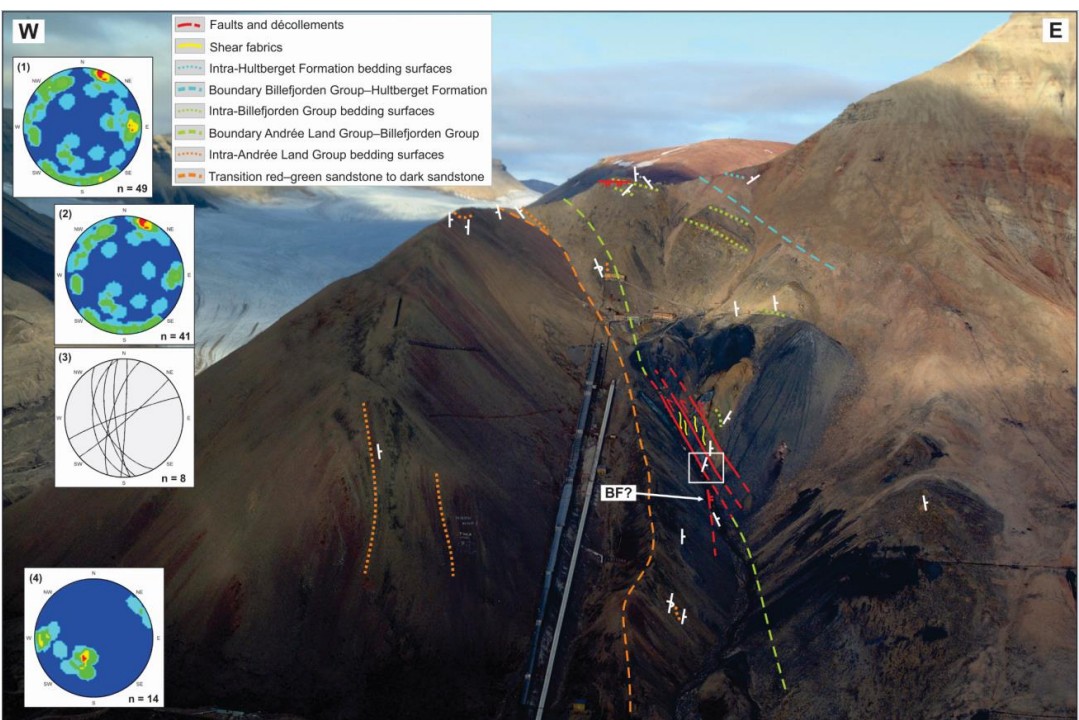

**Figure 2: Aerial photograph of the Pyramiden coal mine locality showing intensely folded (dotted orange lines) Lower Devonian rocks in the west juxtaposed against clastic- and coal-rich, uppermost Devonian–Mississippian sedimentary rocks of the Billefjorden Group, which are overlain by Pennsylvanian–lower Permian strata of the Gipsdalen Group in the east. Dashed lines represent lithostratigraphic transitions. Dotted lines represent bedding surfaces as seen on the photograph, whereas white symbols indicate bedding trend and dip in map view (see Figure 3a). Note the Z-shaped fabrics of uppermost Devonian–Mississippian sedimentary strata (yellow lines) along potential bedding-parallel décollements (red lines) near the boundary between Lower Devonian and uppermost Devonian–Mississippian sedimentary rocks. The white frame shows the location of Figure 3b. Lower hemisphere Schmidt stereonets show (1) contoured poles of fracture surfaces in the uppermost Devonian–Mississippian Billefjorden Group (red indicates high and blue low density), (2) contoured poles of fracture surfaces within sandstone units of the Billefjorden Group, (3) great circles of fracture surfaces within coaly shale- and coal-bearing units of the Billefjorden Group, and (4) contoured poles of fracture surfaces in Lower Devonian rocks. Photo by Åsle Strøm.**






**Figure 3:** **(a) Satellite photograph of the Pyramiden locality (Figure 2) from**
**toposvalbard.npolar.no. See legend in Figure 2. Bedding surface measurements are shown in**
**white. The lower hemisphere Schmidt stereonet shows bedding surface measurements in the**
**Billefjorden Group as contoured poles (red indicates high and blue low density); (b) Field**
**photograph of the base of uppermost Devonian–Mississippian, coaly shale- and coal-rich**
**sedimentary rocks of the Billefjorden Group below the mine entrance in Pyramiden. The**
**photo shows gently east-dipping stratigraphic unit boundaries that localized the formation**
**of bedding-parallel décollements (thick red and thick dashed red lines). Within induvidual**






units, coal displays phyllitic, Z-shaped shear fabrics (yellow lines) parallel–subparallel to
steeply east-dipping, intra-unit bedding surfaces (dashed yellow lines) that are truncated by
subparallel, steeply east-dipping thrusts (thin dashed red lines). See supplement 4 for
uninterpreted photograph. Location is shown in Figure 2; (c) Field photograph showing
gently south-dipping Lower Devonian rocks of the Wood Bay Formation unconformably
overlain by flat-lying strata of the Billefjorden Group (dashed green bedding surfaces),
Hultberget Formation (dashed red bedding surfaces) and Wordiekammen Formation
(dashed blue bedding surfaces) in the hanging wall of the Triungen–Grønhorgdalen Fault
Zone in Triungen (see Figure 1b for location). The upper right inset displays the angular
unconformity (dotted yellow line) between gently south-dipping (tilted?) Lower Devonian
sedimentary rocks of the Wood Bay Formation (bedding surfaces in dashed orange) and
overlying flat-lying strata of the Billefjorden Group; (d) Field photograph of the inferred
location of the Triungen–Grønhorgdalen Fault Zone in Triungen showing that the fault trace
is not exposed and is covered by local black screes probably belonging to uppermost
Devonian–Mississippian coals of the Billefjorden Group. View is towards the west-northwest;
(e) Same as (d) with view towards the north.



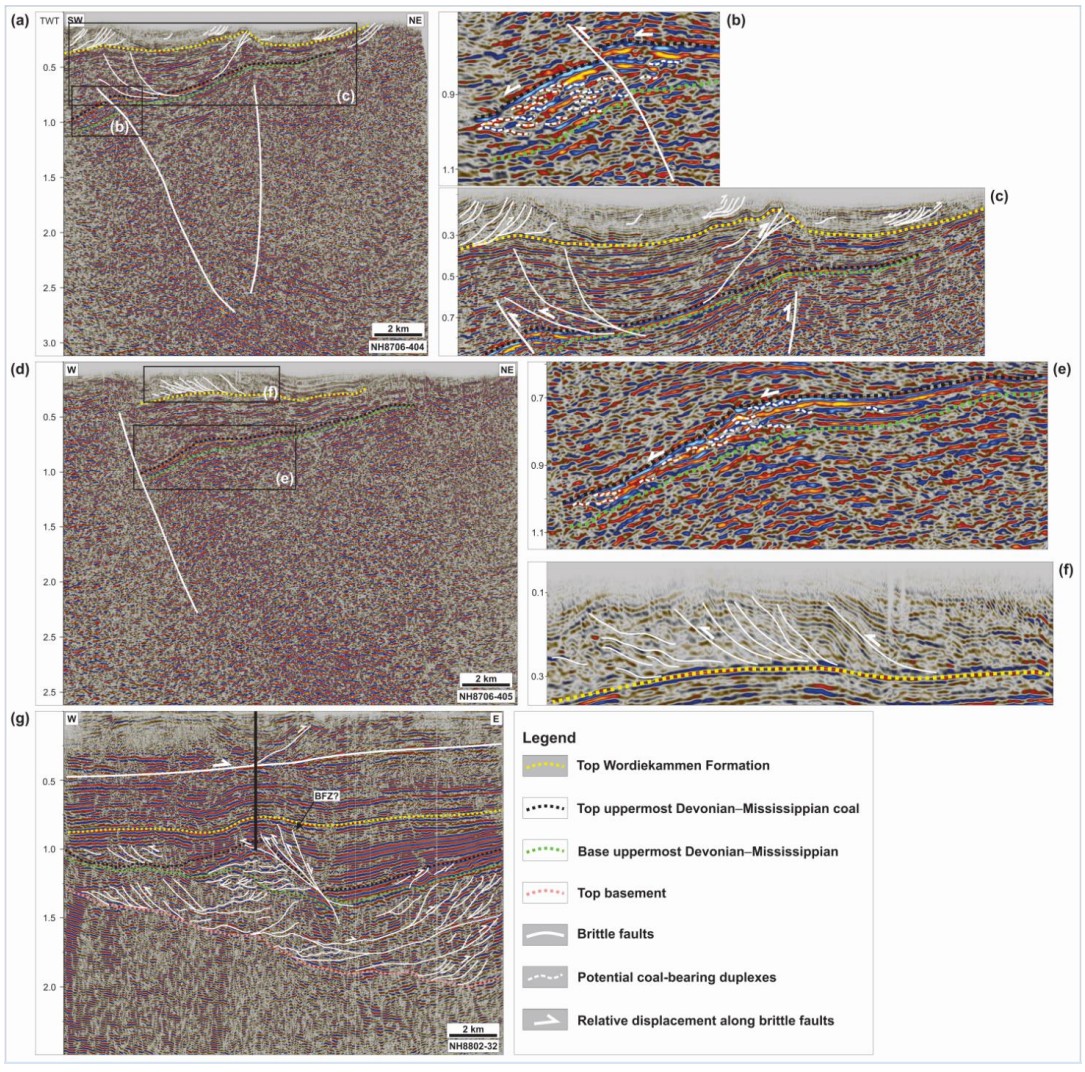

**Figure 4:** **Seismic sections in Two-Way Time (TWT) and associated zoomed-in portions in Saassenfjorden–Tempelfjorden (a–f) and Reindalspasset (g). See Figure 1b for locations. (a) NE–SW-trending section showing minor reverse offset and fault-propagation folding in thickened uppermost Devonian–Mississippian sedimentary rocks along WNW–ESE- to NW–SE-striking, deep-seated basement faults, and Eurekan thrusts in overlying Pennsylvanian–**

**Permian strata; (b) Zoom in SW-verging, coal-bearing duplexes acting as top-SW Eurekan décollements in thickened, uppermost Devonian–Mississippian sedimentary deposits; (c) Zoom in NW–SE-striking Eurekan thrusts that flatten into décollements within uppermost Devonian–Mississippian coals and at the top of the Wordiekammen Formation; (d) NE–west-trending, arch-shaped section showing the potential continuation of the Billefjorden Fault**

**Zone bounding thick uppermost Devonian–Permian sedimentary deposits and top-west Eurekan thrusts within lower Permian rocks; (e) Zoom in coal-bearing duplexes in uppermost Devonian–Mississippian sedimentary strata indicating top-west early Cenozoic**





movement; **(f)** Zoom in Eurekan thrusts flattening into a décollement near the top of the Wordiekammen Formation; **(g)** West–east-trending section showing a top-east Eurekan detachment in Mesozoic sedimentary rocks and a broad anticline in Devonian–Permian strata. Shale-rich Devonian–Mississippian sedimentary strata thicken into the anticline whereas Pennsylvanian–Permian sedimentary rocks thicken away from the anticline. The former are truncated by numerous early Cenozoic Eurekanthrusts arranged into duplex-like structures that flatten into intra Devonian–Mississippian décollements. Note that the thick vertical black line represents the location of exploration well 7816/12-1 (Eide et al., 1991). Abbreviations: BFZ: Billefjorden Fault Zone.







**Figure 5:** **Schematic cross-sections showing the possible evolution of the Billejorden Fault Zone and its relationship with deformation within the Billefjorden Group in Devonian–early Cenozoic times. (a) Relatively high pre-Devonian basement in the east and collapse basin in the west with erosion-related exhumation of basement rocks in the footwall of the Odellfjellet Fault, (b) widespread latest Devonian–Mississippian normal faulting and localization of thick coal deposits on basin edges, (c) latest Mississippian–Pennsylvanian normal faulting localized along the Billefjorden Fault Zone (Balliolbreen and Odellfjellet faults), (d) inversion of Devonian–Carboniferous normal faults and basins during early Cenozoic Eurekan deformation, including top-west thrusting of Proterozoic basement onto Lower–lowermost Upper Devonian rocks of the Andrée Land Group and Mimerdalen Subgroup along the Balliolbreen Fault, the formation of bedding-parallel décollements and thrusts in uppermost Devonian–Mississippian coal-rich deposits, and intense internal deformation of Lower– lowermost Upper Devonian sedimentary rocks (not detailed here) that acted as a buttress, and (e) parts of the cross-section in (d) that fit field observations (from the present contribution and by previous workers) in key localities discussed in the text (Figure 1a–b). Abbreviations: BaF: Balliolbreen Fault; OF: Odellfjellet Fault; TGFZ: Triungen– Grønhorgdalen Fault Zone.**



**Appendix A: List of digitized publications from the Norwegian Polar Institute's Library.**

Abakumov, S. A.: The Lower Hecla Hoeck of the Ny Friesland Peninsula, in: Geology of Spitsbergen, Vol. 1, edited by: Sokolov, V. N., translated by Bradley, Dr. J. E. in 1970, 98–115, 1965.

Andresen, A., Bergh, S. G., Haremo, P., Maher Jr., H. and Welbon, A.: Extrem strain partitioning during evolution of a transform plate boundary, Spitsbergen, North Atlantic, 1992, unpublished.

Birkenmajer, K.: Course of the geological investigations of the Hornsund area, Vestspitsbergen, in 1957–1958, Studia Geologica Polonica, 4, 7–35, 1960.

Birkenmajer, K.: Course of the geological investigations of the Hornsund area, Vestspitsbergen, in 1959 and 1960, Studia Geologica Polonica, 11, 7–33, 1964a.

Birkenmajer, K.: Devonian, Carboniferous and Permian formations of Hornsund, Vestspitsbergen, Studia Geologica Polonica, 11, 47–123, 1964b.

Birkenmajer, K.: Cambrian succession in South Spitsbergen, Studia Geologica Polonica, 59, 7–46,
1978a.

Birkenmajer, K.: Ordovician succession in South Spitsbergen, Studia Geologica Polonica, 59, 47–81, 1978b.

Birkenmajer, K.: Palaeotransport and source of Early Carboniferous fresh-water clastics of South Spitsbergen, Studia Geologica Polonica, 60, 39–43, 1979.

Birkenmajer, K.: Tertiary tectonic deformation of Lower Cretaceous dolerite dykes in a Precambrian terrane, South-West Spitsbergen, Studia Geologica Polonica, 59, 31–44, 1986.

Birkenmajer, K.: Precambrian succession at Hornsund, South Spitsbergen: A lithostratigraphic guide, Studia Geologica Polonica, 98, 7–66, 1992.

Birkenmajer, K. and Morawski, T.: Dolerite intrusions of Wedel-Jarlsberg Land Vestspitsbergen,
Studia Geologica Polonica, 4, 103–123, 1960.

Birkenmajer, K. and Narebski, W.: Precambrian amphibolite complex and granitization phenomena in Wedel-Jarlsberg Land, Vestspitsbergen, Studia Geologica Polonica, 4, 37–82, 1960.

Birkenmajer, K. and Wojciechowski, J.: On the age of ore-bearing veins of the Hornsund area,
Vestspitsbergen, Studia Geologica Polonica, 11, 179–184, 1964.





Burov, Yu. P.: Peridotite inclusions and bombs in the trachybasalts of Sverre Volcano in Vestpitsbergen, in: Geology of Spitsbergen, Vol. 2, edited by: Sokolov, V. N., translated by Bradley, Dr. J. E. in 1970, 267–279, 1965.

Burov, Yu. P. and Livshits, Yu. Ya.: Poorly differentiated dolerite intrusions in Spitsbergen, in: Geology of Spitsbergen, Vol. 2, edited by: Sokolov, V. N., translated by Bradley, Dr. J. E. in 1970, 255–266, 1965.

Burov, Yu. P. and Murashov, L. G.: Some results of the lithological and stratigraphic study of the Kapp Kjeldsen series in the Bockfjorden area, in: Material on the geology of Spitsbergen, edited by: Sokolov, V.N., NIIGA, Leningrad (English translation: The British Library, Lending Division, 1977), 89–97, 1977.

Cerny, J., Lipien, G., Manecki, A. and Piestrzynski, A.: Geology and ore-mineralization of the Hecla Hoek succession (Precambrian) in front of Werenskioldbreen, South Spitsbergen, Studia Geologica Polonica, 98, 67–113, 1992a.

Cerny, J., Plywacz, I. and Szubala, L.: Siderite mineralization in the Hecla Hoek succession (Precambrian) at Strypegga, South Spitsbergen, Studia Geologica Polonica, 98, 153–169, 1992b.

Diβmann, B. and Grewing, A.: Post-svalbardische kompressive Strukturen im westlichen Dickson Land (Hugindalen), Zentral-Spitzbergen, Münster. Forsch. Geol. Paläont., 82, 235–242, 1997.

Firsov, L. V. and Livshits, Yu. Ya.: Potassium–Argon dating of dolerites from the region of Sassenfjorden, Vestspitsbergen, in: Material on the geology of Spitsbergen, edited by: Sokolov, V.N., NIIGA, Leningrad (English translation: The British Library, Lending Division, 1977), 228–234, 1965.

Greving, S., Werner, S. and Thiedig. F.: Post-keldonische Ganggesteine auf der nordöstlichen Mitrahalvøya (Albert I Land, Nordwest-Spitzbergen), Münster. Forsch. Geol. Paläont., 82, 73–78, 1997.

Guddingsmo, J.: Strukturgeologisk analyse av tertiært deformerte karbon/perm-bergater ved Svartfjella, nordvestlige Oscar II Land, Spitsbergen, Master's Thesis, University of Tromsø, Tromsø, Norway, 150 pp.

Haczewski, G.: Lower Carboniferous alluvial sandy deposits (Hornsundneset Formation) of South Spitsbergen, Studia Geologica Polonica, 80, 91–97, 1984.





Haremo, P.: Geological map of the area between Kjellstrømdalen and Adventdalen/Sassendalen, central Spitsbergen, in: Post – Paleozoic tectonics along the southern part of the Billefjorden and Lomfjorden fault zones and their relation to the west Spitsbergen foldbelt, edited by: Haremo, P. (1992), 1989.

Haremo, P.: Post – Paleozoic tectonics along the southern part of the Billefjorden and Lomfjorden fault zones and their relation to the west Spitsbergen foldbelt, Ph.D. Thesis, University of Oslo, Oslo, Norway, 135 pp., 1992.

Haremo, P., Andresen, A. and Dypvik, H.: Mesozoic extension versus Tertiary compression along the Billefjorden Fault Zone south of Isfjorden, central Spitsbergen, 1993, unpublished.

Kempe, M., Niehoff, U., Piepjohn, K. and Thiedig, F.: Kaledonische und Svalbardische Entwicklung im Grundgebirge auf der Blomstrandhalvøya, NW-Spitsbergen, Münster. Forsch. Geol. Paläont., 82, 121–128, 1997.

Kieres, A. and Piestrzynski, A.: Ore-mineralization of the Hecla Hoek succession (Precambrian) around Werenskioldbreen, South Spitsbergen, Studia Geologica Polonica, 98, 115–151, 1992.

Klubov, B. A.: The main features of the geological structure of Barentsøya, in: Geology of Spitsbergen, Vol. 1, edited by: Sokolov, V. N., translated by Bradley, Dr. J. E. in 1970, 89–97, 1965.

Klubov, B. A., Alekseeva, A. B. and Drozdova, I. N.: On the Triassic coals of Spitsbergen, in: Material on the geology of Spitsbergen, edited by: Sokolov, V.N., NIIGA, Leningrad (English translation: The British Library, Lending Division, 1977), 219–227, 1977.

Krasil'shchikov, A. A.: Some aspects of the geological history of North Spitsbergen, in: Geology of Spitsbergen, Vol. 2, edited by: Sokolov, V. N., translated by Bradley, Dr. J. E. in 1970, 32–48, 1965.

Lamar, D. L., Reed, W. E. and Douglass, D. N.: Structures bearing on the sense and magnitude of displacement and tectonic significance of Billefjorden Fault Zone, Dicksonland, Spitsbergen, Svalbard: Progress report, 1982 field season, Lamar-Merifield, Geologists, Technical report 82-6, 48 pp., 1982.

Lange, M. and Hellebrandt, B.: Geologie, Petrographie und Tektonik des südwestlichen Haakon VII Landes, Nordwest-Spitsbergen, Münster. Forsch. Geol. Paläont., 82, 99–119, 1997.





Lange, M., Hellebrandt, B., Piepjohn, K., Saalmann, K. and Donath, H.-J.: Münstersche Forschungen zur Geologie und Palëontologie, Beiträge zur geologischen Evolution Nordwest-Spitzbergen, 82, 242 pp., 1997.

Laptas, A.: Sedimentary evolution of Lower Ordovician carbonate sequence in South Spitsbergen, Studia Geologica Polonica, 89, 7–30, 1986.

Litjes, B. and Thiedig, F.: Geologie und Petrographie des kristallinen Basements und des paläozoischen Bulltinden Konglomerats am Südufer des St. Jonsfjords (Oscar II Land, NW-Spitzbergen), Münster. Forsch. Geol. Paläont., 82, 165–174, 1997.

Livshits, Yu. Ya.: Tectonic of central Vestspitsbergen, in: Geology of Spitsbergen, Vol. 1, edited by: Sokolov, V. N., translated by Bradley, Dr. J. E. in 1970, 59–75, 1965a.

Livshits, Yu. Ya.: Paleogene deposits of Nordenskiöldbreen Land, Vestspitsbergen, in: Geology of Spitsbergen, Vol. 2, edited by: Sokolov, V. N., translated by Bradley, Dr. J. E. in 1970, 193–215, 1965b.

McCann, A. J.: The Billefjorden Fault Zone, Dickson Land, Svalbard: Basement fault control on cover deformation, Ph.D. Thesis, Imperial College, London, UK, 1993.

Michaelsen, B.: Strukturgeologie des svalbardischen Überschiebungs- und Faltengürtels im zentralen, östlichen Dickson Land, Spizbergen (Structural geology of the Svalbardian fold-and-thrust belt in central–eastern Dickson Land, Spitsbergen), Master's Thesis, University of Münster, Münster, Germany, 134 pp., 1998.

Michaelsen, B., Piepjohn, K. and Brinkmann, L.: Struktur und Entwicklung der svalbardischen MImerelva Synkline im zentralen Dickson Land, Spitzbergen, Münster. Forsch. Geol. Paläont., 82, 203–214, 1997.

Peletz, G., Greving, S. and Thiedig, F.: Der tektonische Bau des überschiebungsgürtels auf der Mitrahalvøya, Albert I Land, NW-Spitzbergen, Münster. Forsch. Geol. Paläont., 82, 79–86, 1997.

Piepjohn, K.: Geological Map of Woodfjorden Area (Haakon VII Land, Andrée Land), NW-Spitsbergen, Svalbard, Scale 1 : 150 000, Fachhochschule Karlsruhe, Department of Surveying and Cartography, 1992.

Piepjohn, K.: Geologische Karte Germaniahalvøya, Haakon VII Land Spitzbergen (Svalbard), Scale 1 : 50 000, Fachhochschule Karlsruhe, 1993.



Piepjohn, K.: Überblick über die Arktis-Expeditionen der Spitzbergen-Arbeitsgruppe von Prof. Dr. F. Thiedig, Geologisch-Paläontologisches Institut der Universität Münster, Münster. Forsch. Geol. Paläont., 82, 1–14, 1997a.

Piepjohn, K.: Erläuterungen zur Geologischen Karte 1:150.000 des Woodfjorden-Gebietes (Haakon VII Land, Andrée Land), NW-Spitzbergen, Svalbard, Münster. Forsch. Geol. Paläont., 82, 15–37, 1997b.

Piepjohn, K. and Thiedig, F.: Erläuterungen zur Geologischen Karte 1:50.000 der Germaniahalvøya, Haakon VII Land, Spitzbergen (Svalbard), Münster. Forsch. Geol.
Paläont., 82, 39–52, 1997a.

Piepjohn, K. and Thiedig, F.: Geologisch-tektonische Evolution NW-Spitzbergens im Paläozoikum, Münster. Forsch. Geol. Paläont., 82, 215–233, 1997b.

Piepjohn, K., Greving, S., Peletz, G., Thielemann, T., Werner, S. and Thiedig, F.: Kaledonische und svalbardische Entwicklung im kristallinen Basement auf der Mitrahalvøya, Albert I
Land, NW-Spitzbergen, Münster. Forsch. Geol. Paläont., 82, 53–72, 1997a.

Piepjohn, K., Brinkmann, L., Diβmann, B., Grewing, A., Michaelsen, B. and Kerp, H.: Geologische und strukturelle Entwicklung des Devon im zentralen Dickson Land, Spitzbergen, Münster. Forsch. Geol. Paläont., 82, 175–202, 1997b.

Roy, J.-C.: La saga des vieux grès rouges du Spitzberg (archipel du Svalbard, Arctique): Une
histoire géologique et naturelle, Charenton-le-pont: Auto-Edition Roy-Poulain, 290 pp., 2009.

Roy, J.-C., Chorowicz, J., Deffontaines, B., Lepvrier, C. and Tardy, M.: Clues of gravity sliding tectonics at the Eifelian–Givetian boundary in the Old Red Sandstone of the [late Silurian?]-Devonian trough of Andrée Land (Spitsbergen), in: La saga des vieux grès rouges du
Spitzberg (archipel du Svalbard, Arctique): Une histoire géologique et naturelle, edited by: Charenton-le-pont: Auto-Edition Roy-Poulain, Norw. J. Geol., unpublished.

Rozycki, S. Z.: Geology of the north-western part of Torrell Land, Vestspitsbergen, Studia Geologica Polonica, 2, 4–98, 1959a.

Rozycki, S. Z.: Geological cross-sections of the north-western part of Torrell Land,
Vestspitsbergen, 1 : 25000, Studia Geologica Polonica, 2, 1959b.

Rozycki, S. Z.: Geological map of the north-western part of Torrell Land, Vestspitsbergen, 1 : 50000, Studia Geologica Polonica, 2, 1959c.



Saalmann, K. and Brommer, A.: Stratigraphy and structural evolution of eastern Brøggerhalvøya, NW-Spitsbergen, Münster. Forsch. Geol. Paläont., 82, 147–164, 1997.

Saalmann, K., Piepjohn, K. and Thiedig, F.: Involvierung des Tertiärs von Ny-Ålesund in den alpidischen Deckenbau der Brøggerhalvøya, NW-Spitzbergen, Münster. Forsch. Geol. Paläont., 82, 129–145, 1997.

Siedlecki, S.: Culm beds of the SW. coast of Hornsund, Vestspitsbergen, Studia Geologica Polonica, 4, 93–102, 1960.

Siedlecki, S. and Turnau, E.: Palynological investigations of culm in the area SW of Hornsund, Vestspitsbergen, Studia Geologica Polonica, 11, 125–140, 1964.

Smulikowski, W.: Petrology and Some Structural Data of Lower Metamorphic Formations of the Hecla Hoek Succession in Hornsund, Vestspitsbergen, Studia Geologica Polonica, 18, 3–107, 1965a.

Smulikowski, W.: Geological sketch-map of upper Revdalen, 1 : 2500, Studia Geologica Polonica, 18, 1965b.

Smulikowski, W.: Geological map of Kvartsittsletta and of SW margin of Werenskioldbreen, 1 : 5000, Studia Geologica Polonica, 18, 1965c.

Smulikowski, W.: Directions of linear structures in the Hecla Hoek succession of the SW Wedel-
Jarlsberg Land, 1 : 25000, Studia Geologica Polonica, 18, 1965d.

Smulikowski, W.: Sketch-map to show areas of detailed petrological and structural investigations in the Hecla Hoek succession of the SW Wedel-Jarlsberg Land, 1 : 25000, Studia Geologica Polonica, 18, 1965e.

Smulikowski, W.: Some petrological and structural observations in the Hecla Hoek succession
between Werenskioldbreen and Torellbreen, Vestspitsbergen, Studia Geologica Polonica, 21, 97–161, 1968a.

Smulikowski, W.: Geological map of the environs of Werenskioldbreen, 1 : 25000, Studia Geologica Polonica, 21, 1968b.

Thielmann, T. and Thiedig, F.: Paläozoisch-postkaledonische Sedimente auf Mitrahalvøya, NW-
Spitzbergen, Münster. Forsch. Geol. Paläont., 82, 87–98, 1997.

Ustritskii, V. I.: Main features of the stratigraphy and palaeogeography of the upper Palaeozoic of Spitsbergen, in: Material on the geology of Spitsbergen, edited by: Sokolov, V.N., NIIGA,





Leningrad (English translation: The British Library, Lending Division, 1977), 98–124, 1967.

Wojciechowski, J.: Ore-bearing veins of the Hornsund area, Vestspitsbergen, Studia Geologica Polonica, 11, 173–177, 1964.

Witt-Nilsson, P. W.: The West Ny Friesland Terrane: An Exhumed Mid-Crustal Obliquely Convergent Orogen, Ph.D. Thesis, Uppsala University, Uppsala, Sweden, 121 pp., 1998.

Witt-Nilsson, P. W.: Caledonian mid-crustal oblique convergence in eastern Svalbard, 34 pp.,
1998, unpublished.

Witt-Nilsson, P. W., Hellmann, F. J., Johansson, Å, Larionov, A. N. and Tebenkov, A. M.: Structural and geochronological studies of mylonites along a major Caledonian fault zone, northeastern Spitsbergen, 36 pp., 1998, unpublished.

Wright, N. J. R.: The Billefjorden Group Central and Eastern Spitsbergen, Cambridge Arctic Shelf
Program Report, 6, 79 pp., 1975a.

Wright, N. J. R.: The Billefjorden Group of Western Spitsbergen, Cambridge Arctic Shelf Program Report, 9, 37 pp., 1975b.

Wright, N. J. R.: The Carboniferous and Permian evolution of Svalbard, Cambridge Arctic Shelf Program Report, 25, 51 pp., 1976.