# Peer review of "Early Cenozoic Eurekan strain partitioning and decoupling in central Spitsbergen, Svalbard"

_Solid Earth, 2020_

## Referee Comment (RC1) · Anonymous Referee #1 · 5 Jan 2021

General comments The manuscript titled "Early Cenozoic Eurekan strain partitioning and decoupling in central Spitsbergen, Svalbard" by Jean-Baptiste P. Koehl combines seismic data with field and petrographic observations to address the deformation patterns in the Devonian-Permian sedimentary successions in central Spitsbergen. This study shows that during the Devonian there was not reverse movement, and instead the region underwent experienced normal faulting during the Carboniferous followed by early Cenozoic reactivation and fold-thrust belt formation. Overall, the paper is a very interesting work presenting data that contributes to scientific knowledge of the Spitsbergen area. However, some updates are still needed regarding the interpretation of the seismic data as well as the overall organization of the text. Specific comments Text organization and clarifications: (Line 45): The introduction does not state the signifi-

cance of this study. It reads more as an outline of the paper rather than introduction to the problem. Each paragraph starts with the phrase "this study...". For someone who is not interested or not familiar with the study area it is hard to follow. I would recommend briefly summarizing 1) previous work and lack of knowledge, 2) broader impact, 3) questions that the paper addresses 4) and methods that were used to address the questions. After reading the introduction of this paper, the reader would expect just a case study for the Spitsbergen area, however, this study has a broader impact for fold and thrust systems worldwide, and this needs to be clear in the introduction.

(Line 206): In the methodology section, I would recommend separating the three different methodologies into different sections. The field and petrologic methodology sections lack significant information about the methodological steps that the author performed. In which lithologies were the data measured? Where is the raw data presented (which are in figure 2,3) etc? Additionally, remote satellite imagery is not mentioned in the methodology section neither in the discussion.

In the "Results" session a lot of parts include interpretations, comparisons and discussion about this study and previous studies. These remarks do not belong in the results section and they should be in the discussion. Alternatively, the author could create a new "interpretation section". Examples in lines: 231, 239, 256, 267-271, 290-299, 326, 402

(Line 235): What is the exact lithology of these sedimentary rocks? Sample numbers?

Context and major issues As mentioned above, the motivation and broader impact of this work is not clear from the beginning in the abstract and introduction.

Line 326: Is that information coming from the well—it is unclear! How can you tell what type of lithology it is just from seismic reflection data? As stated before, this part is an interpretation and not part of the results section (raw description of reflectors). In the discussion you mention that this information comes from the well, but it should clearly be stated in the results (i.e. core, cuttings, well logs?).

Need an extra figure: I believe a figure showing the stratigraphy will be very useful. Especially for a non-expert in the area it is hard to follow the different terminology and complicated formation names. The author could recycle and modify a pre-existing stratigraphic model ex. figure 2 from Piepjohn and Dallmann 2014.

I have some major issues with the interpretation of the seismic data. A lot of the seismic images are significantly overinterpreted (drawing lines blindly over non-existent features which are poorly imaged) and some of the labeled "brittle faults" do not appear in a realistic geometry nor do they have fault-plane reflections or offset adjacent reflections with enough confidence given the quality of the data. Example in figure 4b – the author connects positive to negative reflection peaks. In figure 4g a lot of the brittle faults below the base of Devonian Mississippian cross cuts reflections in strange patterns and they are not well imaged. I would recommend revisiting the seismic interpretations and removing overinterpreted lines, since they are misleading. The author should focus on the major structures which are nicely imaged in Figure 4, and not overinterpret the deeper section where the resolution of the data is poor (e.g. Figure 4g – unnecessary lines and not convincing interpretations between pink and green horizons).

(Line 720): I agree that the map on this part is wrong – please use some field photos. Where does the crystalline basement actually appears?

In figure 5 the illustration of the large displacement of the basement is not supported by the seismic lines neither is explained in the text. Figure 5 needs to have a vertical scale associated with it. It appears that multiple kilometers of reverse basement throw are drawn across the BaF – where does this come from? The seismic images in Figure 4 support only modest reactivation of older normal fault features.

Figures Figure 4: The color scheme is not color blind friendly, the green lines on top of blue and red will be impossible for some people to see. Please consider changing the seismic colors. The legend is not very clear – please reword. "top uppermost

Devonian-Mississippian coal". Again, this would be clearer if the author included a stratigraphic column. Add in the legend the well for 4g. What is the length of the well? Please add a scale.

Figure 5: (E) the schematics are very small. I would recommend re-arranging this figure. Can you make them all the same size? Or maybe put boxes on the final version of D?

Figure 6 and 7 in the supplementary material are not discussed or mentioned anywhere in the main text.

Technical corrections: Missing references: Lines 223-231 Line 310: "are believed to represent", missing reference! Add figure calls line 338 Grammar and syntax mistakes: The text still needs some work for grammar and syntax mistakes. A general suggestion is to shorten sentences – there are sentences 10 lines long with a lot of commas and parentheses that makes it hard to read and follow. A few examples of long sentences, and grammar or syntax mistakes: In the abstract: First sentence is 9 lines, Line 86, Line 89: Caledonian grain, Line 97, Line101, Line 108, Line 492, Line 494, Line 576, Line 777, Line 799

---

## Referee Comment (RC2) · Thomas Phillips (Referee) · 17 Mar 2021

Solid Earth – 2020-165 Koehl – Early Cenozoic Eurekan strain partitioning and decoupling in Central Spitsbergen, Svalbard See supplement for pdf version of this review.

This paper evaluates aspects of the geological evolution of the Spitsbergen area of Svalbard, combining field- and seismic-based observations. The paper shows how coal-bearing intervals may effectively decouple and partition strain above and below, negating the need to invoke a period of Late Devonian contraction in the area. Furthermore, the paper demonstrates how the major fault in the area, the Billefjorden Fault Zone, did not undergo reverse reactivation previously purported to relate to this Late Devonian event. The paper is very detailed and critically evaluates information from a
large range of previous studies, and will be of great interest to people working in this area. However, the wider implications of the study are currently lacking outside of a local geological context. In addition, there are a number of issues that make the paper difficult to access for someone not already familiar with the area. I here list a series of overall comments below, followed by technical queries and corrections.

1. The introduction is relatively narrow and the overall aims of the study are unclear. At present the introduction outlines that the study aims to achieve, but does not place these into the wider context. It should be made clearer at this point what are the rationale and key scientific questions to be addressed in this study and how does this compare/contrast with previous studies in the area. In addition, it would be good to consider the wider implications of the study, separated from their local context, e.g. examining how deformation may be partitioned across coal-bearing intervals in rift systems generally as opposed to just in this locality. A further interesting aspect that could be expanded upon is the integration of seismic and field observations and the difference in scale between the two. A scale is required on Figure 3b. 2. The stratigraphy and nomenclature used throughout can be difficult to follow and to relate to the figures. A combination of formation names, groups and ages is used throughout the manuscript. I would recommend establishing these early in the manuscript by establishing a stratigraphic framework and including a stratigraphic column for the area as a figure. In addition, it would be worth increasing the annotation on the figures to enable greater cross-referencing between figures and text. This is especially important with regards to the ages of the different intervals on the seismic section and satellite images, and also to identify features when multiple sub-figures are called out simultaneously in the text. 3. Figures – there are currently only 5 figures in the manuscript which are heavily used and referred to in the text. It would be worth including more information on these figures with increased annotation or including new figures, such as the aforementioned stratigraphic column. In particular, it would be worth including some close-ups of the map based figures to show stratigraphic relationships (e.g. L203, 854). Additional figures such as 5e should be expanded and further explained

on the figure itself. 4. The broader implications of the study should be explored in more detail. At present the Discussion focusses on a range of different theories regarding details of the evolution of Spitsbergen. Comparisons and the implications for similar rift systems should be drawn to emphasise the broader implications of this study – e.g. how does this compare to other rift systems where deformation is partitioned. 5. At present, the key points of the paper can be lost in the discussion section discussing the various models and competing ideas for the evolution of various aspects of Spitsbergen geology. This would be made clearer by incorporating more figures related to these models and establishing the stratigraphic framework early in the paper with the aid of a stratigraphic column. However, the discussion still accounts for a large proportion of the overall paper and could be shortened to focus on the key aspects of the paper as outlined in the title and introduction of the paper and backed up by the data shown. The early points of the conclusions (1-4) are succinct and very interesting, however the latter points are less clear from the figures and do not contribute as much to the overall story.

Technical comments Line 82 – When did the orogeny stop? Line 153 – Where is the Billefjorden Fault Zone located? And how does it relate to the Balliobreen Fault and the Odefjellet fault? This is not clear on Figure 1, where the fjord is labelled, but not the fault zone. Also, the text refers to Carboniferous deposits, but the figure to Pennsylvanian. A stratigraphic column would help greatly associated with this figure. L194 – Can you show some indication of the orientation on Figure 1a. It appears to be reflected in the orientation of the fjord and some landscape lineations but this is not clear from the text or figure. L203 – Unconformable relationship is not clear from the figure. Close up of the area would be beneficial. Is the Billefjorden fault zone present on the map? L211 – What is the purpose of the microscopic analyses, is this to confirm structural measurements? Figure 2 – Label the location of the mine entrance, along with other key features referred to in the text (e.g. the different groups and formations) L242 – change to 1-2 m L244 – potentially change to > 3m Figure 4 – Basement horizon not always interpreted on the subfigures Figure 4g – Label the well name on the section.

Duplex interpretation is clear, but wedge-shaped geometries difficult to identify. Figure 4b,e – Z-shaped geometries not immediately clear on the figure. Label on the figure? L400 – difficult to tell what is being referred to L557 – Very long sentence that is difficult to follow. Breakup to make clearer. L583 – Is this an example of where there is no decoupling interval present? If so this should be stated. L601 – State explicitly how this model relates to your observations, is it in agreement? Figure 5 – More labelling is required on the figure, e.g. the collapsing orogen and exhuming core complexes are not present/clear on 5a. L813 – Exposure of the basement is also not shown on the figure?

Please also note the supplement to this comment:
https://se.copernicus.org/preprints/se-2020-165/se-2020-165-RC2-supplement.pdf

---

## Author Comment (AC1) · 22 Mar 2021

Dear Sir, Madam, thank you very much for your input on the manuscript, it is highly appreciated. Here is my reply to your comments. I hope that the changes implemented improve the shortcomings of the manuscript highlighted by your comments and suggestions. Please do not hesitate to contact me shall this not be the case for some comments.

1. Comments from anonymous referee Comment 1: General comments The manuscript titled "Early Cenozoic Eurekan strain partitioning and decoupling in central Spitsbergen, Svalbard" by Jean-Baptiste P. Koehl combines seismic data with field and petrographic observations to address the deformation patterns in the Devonian-

Permian sedimentary successions in central Spitsbergen. This study shows that during the Devonian there was not reverse movement, and instead the region underwent experienced normal faulting during the Carboniferous followed by early Cenozoic reactivation and fold-thrust belt formation. Overall, the paper is a very interesting work presenting data that contributes to scientific knowledge of the Spitsbergen area. However, some updates are still needed regarding the interpretation of the seismic data as well as the overall organization of the text. Comment 2: Specific comments Text organization and clarifications: (Line 45): The introduction does not state the significance of this study. It reads more as an outline of the paper rather than introduction to the problem. Each paragraph starts with the phrase "this study: : :". For someone who is not interested or not familiar with the study area it is hard to follow. I would recommend briefly summarizing 1) previous work and lack of knowledge, 2) broader impact, 3) questions that the paper addresses 4) and methods that were used to address the questions. Comment 3: After reading the introduction of this paper, the reader would expect just a case study for the Spitsbergen area, however, this study has a broader impact for fold and thrust systems worldwide, and this needs to be clear in the introduction. Comment 4: (Line 206): In the methodology section, I would recommend separating the three different methodologies into different sections. Comment 5: The field and petrologic methodology sections lack significant information about the methodological steps that the author performed. Comment 6: In which lithologies were the data measured? Comment 7: Where is the raw data presented (which are in figure 2,3) etc? Comment 8: Additionally, remote satellite imagery is not mentioned in the methodology section neither in the discussion. Comment 9: In the "Results" session a lot of parts include interpretations, comparisons and discussion about this study and previous studies. These remarks do not belong in the results section and they should be in the discussion. Alternatively, the author could create a new "interpretation section". Examples in lines: 231, 239, 256, 267-271, 290-299, 326, 402. Comment 10: (Line 235): What is the exact lithology of these sedimentary rocks? Sample numbers? Comment 11: Context and major issues As mentioned above, the motivation

and broader impact of this work is not clear from the beginning in the abstract and introduction. Comment 12: Line 326: Is that information coming from the wellâ˘Y˘A˘G Tit is unclear! How can you tell what type of lithology it is just from seismic reflection data? As stated before, this part is an interpretation and not part of the results section (raw description of reflectors). In the discussion you mention that this information comes from the well, but it should clearly be stated in the results (i.e. core, cuttings, well logs?). Comment 13: Need an extra figure: I believe a figure showing the stratigraphy will be very useful. Especially for a non-expert in the area it is hard to follow the different terminology and complicated formation names. The author could recycle and modify a pre-existing stratigraphic model ex. figure 2 from Piepjohn and Dallmann 2014. Comment 14: I have some major issues with the interpretation of the seismic data. A lot of the seismic images are significantly overinterpreted (drawing lines blindly over non-existent features which are poorly imaged) and some of the labeled "brittle faults" do not appear in a realistic geometry nor do they have fault-plane reflections or offset adjacent reflections with enough confidence given the quality of the data. Comment 15: Example in figure 4b – the author connects positive to negative reflection peaks. Comment 16: In figure 4g a lot of the brittle faults below the base of Devonian Mississippian cross cuts reflections in strange patterns and they are not well imaged. I would recommend revisiting the seismic interpretations and removing overinterpreted lines, since they are misleading. Comment 17: The author should focus on the major structures which are nicely imaged in Figure 4, and not overinterpret the deeper section where the resolution of the data is poor (e.g. Figure 4g – unnecessary lines and not convincing interpretations between pink and green horizons). Comment 18: (Line 720): I agree that the map on this part is wrong – please use some field photos. Where does the crystalline basement actually appears? Comment 19: In figure 5 the illustration of the large displacement of the basement is not supported by the seismic lines neither is explained in the text. Comment 20: Figure 5 needs to have a vertical scale associated with it. Comment 21: It appears that multiple kilometers of reverse basement throw are drawn across the BaF – where does this come from? The seismic images in Figure 4 support only modest reactivation of older normal fault features. Comment 22: Figures Figure 4: The color scheme is not color blind friendly, the green lines on top of blue and red will be impossible for some people to see. Please consider changing the seismic colors. Comment 23: The legend is not very clear – please reword "top uppermost Devonian-Mississippian coal". Comment 24: Again, this would be clearer if the author included a stratigraphic column. Comment 25: Add in the legend the well for 4g. What is the length of the well? Please add a scale. Comment 26: Figure 5: (E) the schematics are very small. I would recommend re-arranging this figure. Can you make them all the same size? Or maybe put boxes on the final version of D? Comment 27: Figure 6 and 7 in the supplementary material are not discussed or mentioned anywhere in the main text. Comment 28: Technical corrections: Missing references: Lines 223-231. Comment 29: Line 310: "are believed to represent", missing reference! Comment 30: Add figure calls line 338. Comment 31: Grammar and syntax mistakes: The text still needs some work for grammar and syntax mistakes. A general suggestion is to shorten sentences – there are sentences 10 lines long with a lot of commas and parentheses that makes it hard to read and follow. A few examples of long sentences, and grammar or syntax mistakes: In the abstract: First sentence is 9 lines, Line 86, Line 89: Caledonian grain, Line 97, Line101, Line 108, Line 492, Line 494, Line 576, Line 777, Line 799.

2. Author's reply Comment 1: partly agreed. Updates regarding the organization of the text are discussed in the replies to comments 2–13 and updates regarding seismic interpretation are discussed in the replies to comments 14–16. Comment 2: agreed. Comment 3: agreed. Comment 4: agreed. Comment 5: agreed. Comment 6: agreed. Comment 7: the raw data shown in figures 2 and 3 are presented in the method section lines 208–211. Comment 8: agreed. Satellite images were not used in the present study. Comment 9: agreed. Comment 10: agreed. Comment 11: agreed. Comment 12: agreed. Comment 13: agreed. Comment 14: disagreed. The structures interpreted represent major disruption surfaces that may be identified on the high-resolution, uninterpreted version of the seismic sections (provided as a supplement).

Comment 15: disagreed. Since the surfaces mapped represent formation/group tops and bases, they do not necessarily only coincide with vertical lithological transitions but also possibly with horizontal lithological transitions. An example is the coals of the Billefjorden Group, which are found at varying depth onshore in Dickson Land (see Cutbill et al., 1976 their figure 7). Comment 16: disagreed. The shallowest parts of the seismic section in figure 4g are relatively easy to interpret due to well constrain (tie to well 7816/12-1) and sub-horizontal seismic reflections, and have already been interpreted in Bælum and Braathen (2012). The present manuscript focuses on the deeper interval. The author of the present manuscript also disagrees in that the lower part of the seismic section is of low quality. Disruption surfaces forming duplex-like geometries appear clearly when zooming in the high-resolution version of the figure. This is also in the opinion of the other referee Dr. Phillips (see his comment 24). Comment 17: disagreed. See reply to comment 16. Comment 18: agreed. Basement does not crop out. Field photographs do not illustrate this well and, instead, the author of the present manuscript shows microscope photographs on both sides of the fault (supplement S1). Comment 19: disagreed. If the anonymous referee is referring to "large displacement of the basement" in figure 5a, it is explained by Devonian, down-east normal faulting along the Billefjorden Fault Zone (lines 808–812). If the anonymous referee is referring to "large displacement of the basement" in figure 5b–c, it is explained by latest Devonian–Carboniferous, down-east, rift-related normal faulting (lines 812–814). If the anonymous referee is referring to "large displacement of the basement" in figure 5d, it is explained by early Cenozoic, Eurekan inversion of the Balliolbreen Fault segment of the Billefjorden Fault Zone (lines 814–820). Comment 20: disagreed. Figure 5 is a schematic cross section and does not require a scale. Future accurate reconstructions like that of Koehl et al. (2020, their figure 11) in Billefjorden definitely need a scale because they discuss the thickness of stratigraphic units. This is not the case of the present study. Comment 21: agreed. Seismic data support only minor early Cenozoic top-west of the Billefjorden Fault Zone. However, field observations of Proterozoic basement in the hanging wall juxtaposed against Devonian strata in the footwall between Pyramiden and Odellfjellet (see locations in figure 1 in the present manuscript) suggest significant top-west displacement along the fault in the early Cenozoic. These along-strike variations in the amount of early Cenozoic movement are suggested to be related to pobable segmentation of the Billefjorden Fault Zone by sub-orthogonal fault zones that partitioned Eurekan deformation in the early Cenozoic (lines 861–863 and further discussed lines 864–882). Comment 22: agreed. Comment 23: agreed. Comment 24: agreed. Comment 25: partly agreed. There is little space in the figure to add the name of the well and adding one item in the legend for a single feature is probably not appropriate. The caption already states that the thick black line in figure 4g represents the location of the well. Nevertheless, the total depth of the well should be mentioned. Comment 26: agreed. Comment 27: agreed. These figures were part of a previous manuscript and are not part of the present manuscript. A new link to the online open access repository containing exclusively the figures of the present manuscript will be provided instead of the current link. Comment 28: agreed, the sentence is poorly written. Comment 29: agreed, the sentence is not clear. Comment 30: agreed. Comment 31: agreed, though some sentences are not thought to be possible to rephrase or shorten and keep the exact same meaning.

3. Changes implemented Comment 1: see reply to comments 2–16. Comment 2: replaced " discusses the presence of " by "has potential implications for strain partitioning in rift systems and distal parts of fold-and-thrust belts. Notably, the study describes " line 54. Added ", which were, thus far, not described" and "discusses " line 57. Added "Hence, the study contributes to our understanding of deformation partitioning in fold-and-thrust belts consisting of thick sedimentary successions, and for the extent of the Ellesmerian Orogeny in the Arctic, which presumably extends from Arctic Canada and northern Greenland to Spitsbergen." lines 84–87. Changed "Finally, the study has implication for the segmentation and linkage of rift-bounding fault with long-lived tectonic histories. Thus far, although segmentation of the Billefjorden Fault Zone was described (e.g., Bælum and Braathen, 2012), along-strike geometrical and kinematics variations along the Billefjorden Fault Zone have been poorly addressed and tentatively attributed

to the complex tectonic history of this fault. The present study further discusses the significant along-strike variations in geometry and kinematics, the extent, and potential segmentation of the Billefjorden Fault Zone in conjunction with a new trend of NNE-dipping faults striking suborthogonal to the main N–S-trending structural grain in the study area. The role of these suborthogonal faults in Eurekan strain partitioning is briefly discussed." into "Finally, the study has implication for the segmentation and linkage of rift-bounding fault with long-lived tectonic histories. Thus far, although segmentation of the Billefjorden Fault Zone was described (e.g., Bælum and Braathen, 2012), along-strike geometrical and kinematics variations along the Billefjorden Fault Zone have been poorly addressed and tentatively attributed to the complex tectonic history of this fault. The present study further discusses the significant along-strike variations in geometry and kinematics, the extent, and potential segmentation of the Billefjorden Fault Zone in conjunction with a new trend of NNE-dipping faults striking suborthogonal to the main N–S-trending structural grain in the study area. The role of these suborthogonal faults in Eurekan strain partitioning is briefly discussed." lines 88–97. Changed "discusses the presence of " into "has potential implications for strain partitioning in rift systems and distal parts of fold-and-thrust belts. Notably, the study describes " lines 53–54. Added "The identification of structures showing comparable geometries and kinematics (e.g., bedding-parallel décollements) within discrete stratigraphic units (e.g., coals and coaly shales of the Billefjorden Group) both on nearshore seismic data and onshore during structural fieldwork further validates the use of seismic interpretation in areas where extensive (glacial) erosion resulted in partial destruction and covering of outcrop transects with loose material, and where large portions of the outcrops available for field mapping are hardly accessible for detailed inspection because located on steep slopes and cliffs. The study also illustrates the complementarity between fieldwork, which provide detailed lithological and structural data, and seismic transects providing continuous transects through deformation belts and fault zones." lines 61–69 and added a scale to figure 3b. Comment 3: see reply to comment 2. Comment 4: split each method into a paragraph. Comment 5: replaced "structural"

by "strike and dip" line 208, added " These were used to determine the state of deformation of the various lithological units of the Billefjorden Group, to infer the presence of major faults, and to assess fault kinematics. Unfortunately only few slickensides of poor quality were recorded and these are not presented in the present study." lines 210–213, and added "Thin sections were cut perpendicular to potential brittle faults in the field to better observed brittle deformation and offset. Cohesive fault rock was exclusively encountered in a gully below the mine entrance in Pyramiden, along the potential field occurrence of the Balliolbreen Fault." lines 215–218. Additional information about plotting of structural data are included in the caption of figures 2 and 3 (lines 1403–1409 and 1412–1413). Comment 6: replaced "sedimentary strata" by "sandstone, coals, and coaly shales of the Billefjorden Group" lines 209–210. Comment 7: none. Comment 8: deleted the sub-section name line 207. Comment 9: changed the name of the chapter to "Results and interpretations" line 220. Comment 10: added "(quartzitic sandstone) " line 236 and "quartzitic " line 719. Comment 11: see reply to comment 2. Comment 12: replaced "likely composed" by "interpreted to consist" lines 326–327. Comment 13: added a new stratigraphic chart as figure 2. Comment 14: none. Comment 15: none. Comment 16: none. Comment 17: none. Comment 18: none. Comment 19: none. Comment 20: none. Comment 21: none. Comment 22: changed the color of green lines in figure 4 to brighter green. Comment 23: changed "uppermost Devonian–Mississippian" in figure 4's legend to "Billefjorden Group". Comment 24: a stratigraphic column figure was included. Comment 25: added "total depth: 2261 meters;" lines 1447–1448. Comment 26: added boxes in figure 5d to show the location of the schematics in figure 5e and "The location of the schematics in (e) is shown as black frames in (d)." to the figure caption. Comment 27: updated link to online repository for access to high-resolution versions of the figures lines 52, 321, and 929. Comment 28: changed "likely composed" into "interpreted to consist" lines 326–327. Comment 29: changed "believed" into "interpreted" line 310. Comment 30: added reference to new stratigraphic column figure line 338. Comment 31: replaced "shows that" by "describes" line 14, deleted ", which" line 16, and replaced ", " by ". The

study shows that these structures" lines 19–20. Replaced ", e.g.," by ". An example is" line 90–91. Replaced ", thus explaining " by ". This would explain " line 560, ", " by ". These contractional duplexes " lines 562–563, "decoupling" by "decoupled" line 563, and "shielding" by "shielded" line 565. Deleted "Based on field data in Pyramiden and seismic data in Sassenfjorden and Reindalspasset, and on previous work (Harland et al., 1974; Lamar et al., 1982, 1986; McCann, 1993; Lamar and Douglass, 1995), " lines 786–788. Replaced ", and " by ". Later on, " lines 815–816.

Additional revisions by the author of the present manuscript -Added reference to new stratigraphic chart figure lines 124, 128, 178, 182, 195, 199, 252, 266, 273, 349, 357, and 365. -Changed "are" by "were" line 559. -Changed "Koehl et al. submitted" into "Koehl et al., 2020" lines 607, 610, 738, and 929. -Changed "suggest" into "suggests" line 804. -Corrected "province" into "provenance" line 871. -Replaced "-" by " " line 1449.

---

## Author Comment (AC2) · 22 Mar 2021

Dear Dr. Phillips, thank you very much for your input on the manuscript, it is highly appreciated. Here is my reply to your comments. I hope the changes implemented improve the shortcomings of the manuscript highlighted by your comments and suggestions. Please do not hesitate to contact me shall this not be the case for some comments.

1. Comments from Dr. Phillips Comment 1: 1. The introduction is relatively narrow and the overall aims of the study are unclear. At present the introduction outlines that the study aims to achieve, but does not place these into the wider context. It should be made clearer at this point what are the rationale and key scientific questions to be

addressed in this study and how does this compare/contrast with previous studies in the area. Comment 2: In addition, it would be good to consider the wider implications of the study, separated from their local context, e.g. examining how deformation may be partitioned across coal-bearing intervals in rift systems generally as opposed to just in this locality. Comment 3: A further interesting aspect that could be expanded upon is the integration of seismic and field observations and the difference in scale between the two. A scale is required on Figure 3b. Comment 4: 2. The stratigraphy and nomenclature used throughout can be difficult to follow and to relate to the figures. A combination of formation names, groups and ages is used throughout the manuscript. I would recommend establishing these early in the manuscript by establishing a stratigraphic framework and including a stratigraphic column for the area as a figure. Comment 5: In addition, it would be worth increasing the annotation on the figures to enable greater cross-referencing between figures and text. This is especially important with regards to the ages of the different intervals on the seismic section and satellite images, and also to identify features when multiple sub-figures are called out simultaneously in the text. Comment 6: 3. Figures – there are currently only 5 figures in the manuscript which are heavily used and referred to in the text. It would be worth including more information on these figures with increased annotation or including new figures, such as the aforementioned stratigraphic column. Comment 7: In particular, it would be worth including some close-ups of the map based figures to show stratigraphic relationships (e.g. L203, 854). Comment 8: Additional figures such as 5e should be expanded and further explained on the figure itself. Comment 9: 4. The broader implications of the study should be explored in more detail. At present the Discussion focusses on a range of different theories regarding details of the evolution of Spitsbergen. Comparisons and the implications for similar rift systems should be drawn to emphasise the broader implications of this study – e.g. how does this compare to other rift systems where deformation is partitioned. Comment 10: 5. At present, the key points of the paper can be lost in the discussion section discussing the various models and competing ideas for the evolution of various aspects of Spitsbergen geology. This would be

made clearer by incorporating more figures related to these models and establishing the stratigraphic framework early in the paper with the aid of a stratigraphic column. Comment 11: However, the discussion still accounts for a large proportion of the overall paper and could be shortened to focus on the key aspects of the paper as outlined in the title and introduction of the paper and backed up by the data shown. Comment 12: The early points of the conclusions (1-4) are succinct and very interesting, however the latter points are less clear from the figures and do not contribute as much to the overall story. Comment 13: Technical comments Line 82 – When did the orogeny stop? Comment 14: Line 153 – Where is the Billefjorden Fault Zone located? And how does it relate to the Balliobreen Fault and the Odefjellet fault? This is not clear on Figure 1, where the fjord is labelled, but not the fault zone. Comment 15: Also, the text refers to Carboniferous deposits, but the figure to Pennsylvanian. A stratigraphic column would help greatly associated with this figure. Comment 16: L194 – Can you show some indication of the orientation on Figure 1a. It appears to be reflected in the orientation of the fjord and some landscape lineations but this is not clear from the text or figure. Comment 17: L203 – Unconformable relationship is not clear from the figure. Close up of the area would be beneficial. Is the Billefjorden fault zone present on the map? Comment 18: L211 – What is the purpose of the microscopic analyses, is this to confirm structural measurements? Comment 19: Figure 2 – Label the location of the mine entrance, along with other key features referred to in the text (e.g. the different groups and formations) Comment 20: L242 – change to 1-2 m. Comment 21: L244 – potentially change to > 3m. Comment 22: Figure 4 – Basement horizon not always interpreted on the subfigures Comment 23: Figure 4g – Label the well name on the section. Comment 24: Duplex interpretation is clear, but wedge-shaped geometries difficult to identify. Comment 25: Figure 4b,e – Z-shaped geometries not immediately clear on the figure. Label on the figure? Comment 26: L400 – difficult to tell what is being referred to. Comment 27: L557 – Very long sentence that is difficult to follow. Breakup to make clearer. Comment 28: L583 – Is this an example of where there is no decoupling interval present? If so this should be stated. Comment 29: L601 – State

explicitly how this model relates to your observations, is it in agreement? Comment 30: Figure 5 – More labelling is required on the figure, e.g. the collapsing orogen and exhuming core complexes are not present/clear on 5a. Comment 31: L813 – Exposure of the basement is also not shown on the figure?

2. Author's reply Comment 1: agreed. Comment 2: agreed. Comment 3: agreed. Comment 4: agreed. Comment 5: agreed. Comment 6: agreed. Comment 7: these field relationships are described in other studies and would require the addition of specific field photographs that do not add to the manuscript's discussion. Comment 8: partly agreed. The location of the schematics in figure 5e is shown in figure 5d. These schematics are to be directly compared with onshore field studies by other workers at these localities (e.g., Harland et al., 1974; Lamar et al., 1986; Lamar and Douglass, 1995). Comment 9: agreed. Comment 10: agreed. Comment 11: partly agreed. However, the present manuscript describes new structures along a major fault with long-lived tectonic history. Interpretation of these structures, inferring potential formation mechanism(s) and discussing their impact on the tectonic history of Spitsbergen (e.g., non-occurrence of the Ellesmerian Orogeny in central Spitsbergen) may have important implications for future studies and need to be discussed appropriately. In addition, major issues such as the along-strike variations in the geometry and kinematics of well-studied faults like the Billefjorden Fault Zone require extensive review and mention of previous works and uncertainties in order to reconcile all previous observations into a unified model. Comment 12: partly agreed. Point 5 of the discussion suggests that Ellesmerian tectonism is not necessary to explain differential deformation between folded Devonian strata of the Andrée Land Group and Mimerdalen Subgroup and poorly deformed Pennsylvanian–Permian strata of the Gipsdalen Group in central Spitsbergen. This is a crucial importance for future studies that will hopefully re-examine evidence of Ellesmerian tectonism throughout the Arctic and consider these with care. Point 6 is also quite important in that it highlights the large uncertainties surrounding the geometry of the most-studied fault zone in Svalbard, the Billefjorden Fault Zone. Points 5 and 6 of the conclusion therefore contribute to important ongoing

debates about key tectonic features of the archipelago. Comment 13: agreed. Comment 14: the Billefjorden Fault Zone consists of the Balliolbreen and Odellfjellet fault segments, both of which are labelled in figure 1b. Comment 15: agreed. Comment 16: these structures are located in western Spitsbergen, i.e., away from the study area. These are not the main targets of the manuscript and are therefore not necessary to add to figure 1. See include literature for structural maps of western Spitsbergen. Comment 17: agreed. To show such relationship, one would need to add a field photograph, which can be found in the study referred in the sentence (Harland et al., 1974). Reference to the figure is to show the location of Sentinelfjellet. Comment 18: microscopic analyses were used to confirm the absence of Proterozoic basement and the presence of Devonian quartzitic sandstone on both sides of the N–S-striking fault encountered in the field. The implications of these field relationships are further discussed in section 5.3. Comment 19: agreed. Comment 20: disagreed. Solid Earth standards require spelling of number from one to ten. Comment 21: see reply to comment 20. Comment 22: agreed. This is due to the high amounts of uncertainty as to what lies below sedimentary strata of the Billefjorden Group in places (especially in Sassenfjorden; figure 4a and d). Comment 23: the well name is included in the figure caption. Comment 24: agreed. Comment 25: agreed. Comment 26: agreed. Comment 27: agreed. Comment 28: no, it is not. At the Robertsonbreen locality, coals and coaly shales of the Billefjorden Group may also host a décollement as shown by bedding-parallel thrusts between the Billefjorden Group and Wordiekammen Formation (Dissmann and Grewing, 1997 their figure 6). Comment 29: agreed. Comment 30: the collapsing orogen and metamorphic core complexes were not located in Billefjorden but farther west and east from the area shown in figure 5a. This is not clearly stated in the manuscript. Comment 31: agreed. Exhumation did not necessarily occurred during Carboniferous normal faulting. It may also have occurred during Devonian normal faulting, and due to early Cenozoic thrusting and erosion.

3. Changes implemented Comment 1: replaced " discusses the presence of " by "has potential implications for strain partitioning in rift systems and distal parts of fold-and-

thrust belts. Notably, the study describes " line 54. Added ", which were, thus far, not described" and "discusses " line 57. Added "Hence, the study contributes to our understanding of deformation partitioning in fold-and-thrust belts consisting of thick sedimentary successions, and for the extent of the Ellesmerian Orogeny in the Arctic, which presumably extends from Arctic Canada and northern Greenland to Spitsbergen." lines 84–87. Changed "Finally, the study has implication for the segmentation and linkage of rift-bounding fault with long-lived tectonic histories. Thus far, although segmentation of the Billefjorden Fault Zone was described (e.g., Bælum and Braathen, 2012), along-strike geometrical and kinematics variations along the Billefjorden Fault Zone have been poorly addressed and tentatively attributed to the complex tectonic history of this fault. The present study further discusses the significant along-strike variations in geometry and kinematics, the extent, and potential segmentation of the Billefjorden Fault Zone in conjunction with a new trend of NNE-dipping faults striking suborthogonal to the main N–S-trending structural grain in the study area. The role of these suborthogonal faults in Eurekan strain partitioning is briefly discussed." into "Finally, the study has implication for the segmentation and linkage of rift-bounding fault with long-lived tectonic histories. Thus far, although segmentation of the Billefjorden Fault Zone was described (e.g., Bælum and Braathen, 2012), along-strike geometrical and kinematics variations along the Billefjorden Fault Zone have been poorly addressed and tentatively attributed to the complex tectonic history of this fault. The present study further discusses the significant along-strike variations in geometry and kinematics, the extent, and potential segmentation of the Billefjorden Fault Zone in conjunction with a new trend of NNE-dipping faults striking suborthogonal to the main N–S-trending structural grain in the study area. The role of these suborthogonal faults in Eurekan strain partitioning is briefly discussed." lines 88–97. Comment 2: changed "discusses the presence of " into "has potential implications for strain partitioning in rift systems and distal parts of fold-and-thrust belts. Notably, the study describes " lines 53–54. Comment 3: added "The identification of structures showing comparable geometries and kinematics (e.g., bedding-parallel décollements) within discrete stratigraphic units (e.g.,

coals and coaly shales of the Billefjorden Group) both on nearshore seismic data and onshore during structural fieldwork further validates the use of seismic interpretation in areas where extensive (glacial) erosion resulted in partial destruction and covering of outcrop transects with loose material, and where large portions of the outcrops available for field mapping are hardly accessible for detailed inspection because located on steep slopes and cliffs. The study also illustrates the complementarity between fieldwork, which provide detailed lithological and structural data, and seismic transects providing continuous transects through deformation belts and fault zones." lines 61–69 and added a scale to figure 3b. Also added "See blue hammer (c. 40 cm) on the foreground and person (c. 1.75 m) in the background for scales." lines 1439–1440. Comment 4: a figure with a stratigraphic column was added (new figure 2). Comment 5: see replies to comments 19 and 25. Comment 6: see replies to comments 4 and 5. Comment 7: none. Comment 8: added boxes in figure 5d to show the location of the schematics in figure 5e and "The location of the schematics in (e) is shown as black frames in (d)." to the figure caption. Comment 9: see replies to comments 1–3. Comment 10: see replies to comments 4 and 5. Comment 11: deleted "Based on field data in Pyramiden and seismic data in Sassenfjorden and Reindalspasset, and on previous work (Harland et al., 1974; Lamar et al., 1982, 1986; McCann, 1993; Lamar and Douglass, 1995), " lines 786–788. Comment 12: none. Comment 13: added "late Cambrian–Silurian " lines 101–102. Comment 14: none. Comment 15: see reply to comment 4. Comment 16: none. Comment 17: added "see location in " line 222. Comment 18: none. Comment 19: added labels of Andrée Land, Billefjorden, and Gipsdalen groups and of mine entrance in figure 2. Comment 20: none. Comment 21: none. Comment 22: none. Comment 23: none. Comment 24: replaced "wedge-shaped" by "sigmoid-shaped" lines 437, 438, 441, and 533. Comment 25: added labels "Z-shaped reflections" in figure 4b and e. Comment 26: added " within the Gipshuken Formation" line 427. Comment 27: replaced ", thus explaining " by ". This would explain " line 560, ", " by ". These contractional duplexes " lines 562–563, "decoupling" by "decoupled" line 563, and "shielding" by "shielded" line 565. Comment 28: none. Comment 29: added "All these earlier models and observations are in agreement with the model of strain partitioning and decoupling along bedding-parallel décollements and thrusts proposed by the present study in Pyramiden." Lines 634–636. Comment 30: added "in the west and east" line 812. Comment 31: added "or kept " and "relatively close to the surface " line 844.

Additional revisions by the author of the present manuscript -Added reference to new stratigraphic chart figure lines 124, 128, 178, 182, 195, 199, 252, 266, 273, 349, 357, and 365. -Changed "are" by "were" line 559. -Changed "Koehl et al. submitted" into "Koehl et al., 2020" lines 607, 610, 738, and 929. -Changed "suggest" into "suggests" line 804. -Corrected "province" into "provenance" line 871. -Replaced "-" by " " line 1449.